# Examining the competing effects of contemporary land management vs. land cover changes on global air quality

Anthony Y. H. Wong, Jeffrey A. Geddes

Department of Earth and Environment, Boston University, Boston, MA, USA

*Correspondence to*: Jeffrey A. Geddes (jgeddes@bu.edu)

**Abstract.** Our work explores the impact of two important dimensions of land system changes, land use and land cover change (LULCC) and direct agricultural reactive nitrogen ($N_r$) emissions from soils, on ozone ($O_3$) and fine particulate matter ($PM_{2.5}$) air quality over contemporary (1992 to 2014) time scales. We account for LULCC and agricultural $N_r$ emissions changes with consistent remote sensing products and new global emission inventories, respectively, estimating their impacts on global surface $O_3$ and $PM_{2.5}$ concentrations and $N_r$ deposition using the GEOS-Chem global chemical transport model. Over this time period, our model results show that agricultural $N_r$ emission changes cause a reduction of annual mean $PM_{2.5}$ levels over Europe and northern Asia (up to -2.1 μg m$^{-3}$), while increasing $PM_{2.5}$ levels India, China and eastern US (up to +3.5 μg m$^{-3}$). Land cover changes induce small reductions in $PM_{2.5}$ (up to -0.7 μg m$^{-3}$) over Amazonia, China and India due to reduced biogenic volatile organic compounds (BVOC) emissions and enhanced deposition of aerosol precursor gases (e.g. $NO_2$, $SO_2$). Agricultural $N_r$ emission changes only lead to minor changes (up to ± 0.6 ppbv) in annual mean surface $O_3$ level, mainly over China, India and Myanmar. Meanwhile, our model result suggests a stronger impact of LULCC on surface $O_3$ over the time period across South America, the combination of changes in dry deposition and isoprene emissions results in -0.8 to +1.2 ppbv surface ozone changes. The enhancement of dry deposition reduces surface ozone level (up to -1 ppbv) over southern China, eastern US and central Africa. The enhancement of soil NO emission due to crop expansion also contribute to surface ozone changes (up to +0.6 ppbv) over sub-Saharan Africa. In certain regions, the combined effects of LULCC and agricultural $N_r$ emission changes on $O_3$ and $PM_{2.5}$ air quality can be comparable (> 20%) to that of anthropogenic emission changes over the same time period. Finally, we calculate that the increase in global agricultural $N_r$ emissions leads to a net increase in global land area (+3.67 ×10$^6$ km$^2$) that potentially faces exceedance in critical $N_r$ load (> 5 kgN ha$^{-1}$ yr$^{-1}$). Our result demonstrates the impacts of contemporary LULCC and agricultural $N_r$ emission changes on $PM_{2.5}$ and $O_3$ air quality, and the importance of land system changes on air quality over multi-decadal timescales.

## 1 Introduction

The broad term of land use and land cover change (LULCC) encapsulates both the anthropogenic (e.g. agricultural expansion) and natural (e.g. ecological succession) dimensions of terrestrial biome changes (Reick et al., 2013), which alter the physical and ecophysiological properties of the land surface. These perturbations alter the transfer and uptake of air pollutants by ecosystems, and can also have large impacts on the emission of biogenic volatile organic compounds (BVOCs), which play

vital roles in tropospheric ozone ($O_3$) and secondary organic aerosol (SOA) formation (Fu and Tai, 2015; Ganzeveld et al., 2010; Heald and Geddes, 2016; Heald and Spracklen, 2015; Squire et al., 2014; Wu et al., 2012a).

Agricultural activities, in addition to being a large driver of LULCC (e.g. Ellis, 2015; Ellis et al., 2013; Goldewijk et al., 2017; Kaplan et al., 2011), also introduce an enormous amount of reactive nitrogen into the soil (Galloway et al., 2008), which can be emitted into the atmosphere either as oxidized or reduced nitrogen. The reactive nitrogen oxides emitted from soil, $NO_x$ ($\equiv$ $NO + NO_2$), enhance $O_3$ production when volatile organic compounds (VOCs) are relatively abundant (i.e. $NO_x$-limited regimes), but suppresses $O_3$ production when the concentration of VOCs is relatively low (i.e. VOC-limited regimes) (Sillman et al., 1990). Reactive nitrogen also contributes to aerosol formation. Ammonia ($NH_3$) can combine with the nitrate and sulphate ions to form secondary inorganic aerosol, while the emissions of $NO_x$ can oxidize further and contribute to particulate nitrate formation (Ansari and Pandis, 1998). Indeed, agricultural emissions are the dominant global anthropogenic source of $NH_3$ (Hoesly et al., 2018), and identified as a major contributor to global premature mortality due to particulate matter (PM) pollution (Lelieveld et al., 2015). Trends in atmospheric reactive nitrogen also affect nitrogen deposition (e.g. Geddes and Martin, 2017), with potentially negatively impacts on biodiversity (e.g. Bobbink et al., 2010; Payne et al., 2017; WallisDeVries and Bobbink, 2017), or eutrophication of aquatic ecosystems (e.g. Fenn et al., 2003). These ecosystem impacts may contribute to economic loss comparable to the benefits of extra crop output from LULCC and agricultural emissions (Paulot and Jacob, 2014; Sobota et al., 2015).

Even while land cover at a particular location may not change, modifications in human management of the land (e.g. intensification of agriculture, irrigation practices, fertilizer application, selective harvesting) may still be associated with changes in pollutant emission and uptake. An obvious example would be a region where direct agricultural emissions may have changed, without any concomitant changes in land cover. Reducing $NH_3$ emission, particularly from the agricultural sector, has been explored as a potent strategy of controlling PM pollution (Giannadaki et al., 2018; Pinder et al., 2007; Pozzer et al., 2017). Bauer et al. (2016) suggest agricultural emissions are the main source of present-day $PM_{2.5}$ (fine particulates with an aerodynamic diameter less than 2.5 μm) over eastern US, Europe and northern China. However, as anthropogenic $NO_x$ and $SO_2$ emissions are expected to lower in the future, some aerosol formation chemistry is expected to become less sensitive to $NH_3$ emissions.

The potential impacts of LULCC and agricultural emission changes on air quality have been explored previously. To date, this work has focused on future projections in land use (Bauer et al., 2016; Ganzeveld et al., 2010; Hardacre et al., 2013; Heald et al., 2008; Squire et al., 2014; Tai et al., 2013; Wu et al., 2012b), contrasted pre-industrial estimates of land cover and agricultural emissions with present-day conditions (Heald and Geddes, 2016; Hollaway et al., 2017), or are regional in focus (Fu and Tai, 2015; Geddes et al., 2015; Silva et al., 2016). For example, Wu et al. (2012) propose that LULCC induced by climate, $CO_2$ abundance, and agriculture could significantly affect surface $O_3$ in the future mainly through modulating dry

deposition and isoprene emissions. Over more contemporary timescales (e.g. across the last 20-30 years), Fu et al. (2016), Fu and Tai (2015) and Silva et al., (2016) find that LULCC could have impacts $O_3$ and PM air quality over China and Southeast Asia.

Given the large spatial scale of LULCC (e.g. Hansen et al., 2013; Li et al., 2018) and agricultural emission changes (e.g. Crippa et al., 2018; Hoesly et al., 2018; Xu et al., 2019) over recent decades, these two land system changes could be contributing

substantially to global trends in $O_3$ and PM pollution. While changes in land cover and agricultural emissions actually occur simultaneously across the globe, they are rarely considered together in simulations of air quality from chemical transport models. The importance of studying these combined processes at the same time was highlighted by Ganzeveld et al. (2010) in their analysis of air quality impacts from future land use and land cover changes. In this study, for example, opposing effects on $O_3$ were simulated with decreases in tropical forest soil $NO_x$ emissions being compensated by increases in soil NOx

emissions associated with agriculture. Still, this work did not explore the concomitant changes in ammonia emissions that would be expected with the changes in agricultural activity. It remains unclear to what extent LULCC may either amplify or offset the impacts of some of the associated agricultural emission changes, how this may vary regionally, and to what extent these land system impacts may compare to concomitant changes resulting from other direct anthropogenic emissions (e.g. emissions from industrial and transport sectors).


Consistent long-term land records of land cover derived from satellite remote sensing observations and global anthropogenic emission inventories have become readily available. This opens an opportunity for a more holistic and observationally-constrained assessment of the impacts on global $O_3$ and PM air quality from contemporary changes in LULCC and agricultural emissions simultaneously, which has been advocated by Ganzeveld et al. (2010), and a comparison of these with the effects

from direct anthropogenic emissions. In this study, we model the effects of contemporary LULCC and agriculture emissions changes on global surface $O_3$ and $PM_{2.5}$ levels, and gauge their importance relative to changes in other direct anthropogenic emissions over the same period of time. We also highlight the effect of agricultural emissions changes on nitrogen deposition on land ecosystems. Through our chemical transport model predictions, we aim to identify potential global hotspots of contemporary land changes that may be substantially altering trends in air quality and nitrogen deposition.

**2 Method**

To simulate global changes in surface $O_3$ and $PM_{2.5}$ concentrations due to LULCC, agricultural emissions, and direct anthropogenic emissions over 1992 to 2014, we use the GEOS-Chem chemical transport model (version 12.7.0, https://doi.org/10.5281/zenodo.3634864). We choose our timeframe due to the availability of consistent high-resolution remote sensing products (PFT and LAI maps) and concurrent global emission inventories. We define "direct anthropogenic"

and "agricultural" emissions separately in more detail below.

We perform five sets of simulations summarized in Table 1: (1) a "baseline" scenario where land cover, agricultural emissions, and direct anthropogenic emissions are all set to 1992 levels; (2) an "anthropogenic emission" scenario where direct anthropogenic emissions are updated to 2014 levels; (3) an "anthropogenic emissions and land cover change" scenario where anthropogenic emissions remain updated to 2014, with land cover inputs now prescribed based on updated 2014 data; and (4)

an "anthropogenic emissions, land cover, and agricultural emission change" scenario where direct anthropogenic emissions and land cover inputs remain updated to 2014, with agricultural emissions also updated to 2014 levels. To test the chemical sensitivity of our results, (5) is performed where anthropogenic emissions are held at 1992 levels, but land cover change and agricultural emissions are updated to 2014 levels.

The role of direct anthropogenic emission changes can be evaluated by comparing simulation (1) and (2); the additional role

played by land cover changes over this time period is evaluated by comparing simulation (2) and (3); and finally the additional impact of agricultural emission changes is evaluated by comparing simulation (3) and (4). The latter two effects will be the focus of this paper, but we compare these to the role of direct anthropogenic emission changes for context. Since changes in surface ozone and PM$_{2.5}$ should be sensitive to NO$_x$-VOC ratio and availability of NO$_3^-$ and SO$_4^{2-}$ ions, the sensitivity of the effects from land cover change and agricultural emission changes to anthropogenic emission changes can be quantified by

evaluating simulation (5).

We use assimilated meteorological fields from Modern-Era Retrospective analysis for Research and Applications Version 2 (MERRA-2) (Gelaro et al., 2017) to drive GEOS-Chem. All simulations are carried out at 2° latitudes by 2.5° longitudes resolution over the globe, using identical meteorological fields from 2011 to 2014 in order to exclude meteorological variability from the analysis. The output from 2011 is discarded as spin-up. The GEOS-Chem model simulates O$_3$ chemistry with a

comprehensive HO$_x$-NO$_x$-VOC-O$_3$-BrO$_x$ chemical mechanism (Bey et al., 2001; Mao et al., 2013). Gaseous dry deposition follows Wang et al. (1998) and Wesely (1989), while particle deposition follows Zhang et al. (2001). In GEOS-Chem, the surface exchange modules are uni-directional (which implies that the effects of bidirectional exchanges of trace gases are not explicitly modelled). In certain regions for which the CEDS inventory scales the calculated emissions to a regional inventory, the extent of accounting for bidirectional exchange may depend on the underlying assumptions in the regional inventory

modeling. For example, agricultural ammonia emissions from NEI for the United States includes considering bidirectional ammonia exchange modeling from the Community Multiscale Air Quality Modeling System (CMAQ) (U.S. EPA, 2018). However, we cannot comment with certainty how this is treated elsewhere across the globe, so we assume that neglecting bidirectional exchange of ammonia (and other species for which an atmospheric compensation point may exist) introduces some uncertainty in our simulation (which we discuss in a subsequent section). Wet deposition is described by Liu et al. (2001)

with updates from Amos et al. (2012) and Wang et al. (2011, 2014). The recent update from Luo et al., (2019) on wet deposition parameterization is also included to improve model-observation agreement for sulfate-nitrate-ammonium (SNA) aerosol. The thermodynamics and gas-aerosol partitioning of the NH$_3$-H$_2$SO$_4$-HNO$_3$ system is simulated by ISORROPIA II module

(Fountoukis and Nenes, 2007). A simple yield-based secondary organic aerosol (SOA) estimate is also included (Kim et al., 2015). Other types of aerosol represented in the model include sea salt, dust, primary black carbon (BC) and organic carbon (OC). The total $PM_{2.5}$ mass is then calculated at 35% relative humidity for consistency with the measurement standard in US.

We use anthropogenic and agricultural emissions based on the Community Emission Data System (CEDS) inventory (Hoesly et al., 2018), which contains the estimates of anthropogenic $NO_x$, non-methane volatile organic compounds (NMVOCs), CO, BC, OC, $SO_2$ and $NH_3$ emissions harmonized from a wide range of global and regional inventories. In this inventory, emissions are from six major sectors: energy production, industry, transportation, RCO (residential, commercial, other), agriculture, and waste. For this study, "agricultural emissions" specifically refer to $NO_x$ and $NH_3$ emitted from fertilizer application and manure management, which correspond directly to agricultural nitrogen input. We do not consider the changes in agricultural of other trace species (e.g. $CH_4$, $SO_2$, CO). For simplicity, we assume that agricultural emissions from fertilizer application in CEDS represent "above canopy" emissions to the atmosphere (instead of making assumptions about the implicit treatment of canopy reduction over each region). We note that the fertilizer emissions of represent only a fraction of the total agricultural $NH_3$ emissions we are considering here (e.g. which also include livestock operation), so that uncertainty in a canopy reduction will only affect a fraction of the total. Likewise, fertilizer $NO_x$ emissions are small compared to the total soil $NO_x$ emissions (for which canopy reduction is accounted for online in the Hudman et al. (2012) parameterization).

Biogenic volatile organic compound emissions are calculated by Model of Emissions of Gases and Aerosols from Nature (MEGAN) v2.1 (Guenther et al., 2012). Soil $NO_x$ emission follows Hudman et al. (2012), with fertilizer emissions zeroed out to avoid double counting with the agricultural $NO_x$ emission in CEDS inventory. Fire (Global Fire Emissions Database v4.1, Van Der Werf et al., 2017) and lightning (Murray et al., 2012) emissions are held constant at 2014 level.

We use the European Space Agency Climate Change Initiative (ESA CCI) land cover map (Li et al., 2018) to characterize LULCC and drive the biosphere-atmosphere emission fluxes in our simulations. The ESA CCI land cover map is a consistent global annual land cover time series derived from the satellite observations from the AVHRR, MERIS, SOPTVGT and PROBA-V instruments. It has a native spatial resolution of 300 m following the United Nations Land Cover Classification System. Time-consistent land surface characterization also requires leaf area index (LAI) data. We use the Global Land Surface Satellite (GLASS) product (Xiao et al., 2016) (retrieved from http://globalchange.bnu.edu.cn/), which is a global LAI time series combining AVHRR and MODIS observation. 3-year average (1991 – 1993 average LAI for 1992 land cover, 2013 – 2015 average LAI for 2014 land cover) is used as input for LAI to GEOS-Chem to reduce the possible effect of interannual variability.

This satellite-derived land surface characterization on its own is not directly compatible with the input to the vegetation-related modules in GEOS-Chem, thus requires further harmonization (dry deposition, BVOC emissions, soil $NO_x$ emissions), which

is a common problem for simulations involving land change (e.g. Geddes et al., 2016). We first aggregate and process the ESACCI land cover map with the tool and crosswalk table provided with the land cover product to derive the percentage coverage of plant functional type (PFT) at 0.05° resolution, which is the native resolution of GLASS LAI. The dominant surface type can be readily mapped to the 11 deposition surface type in the Wesely dry deposition model. We adopt the

approach of Geddes et al. (2016) to replace roughness length ($z_0$) from assimilated meteorology with that prescribed for each deposition surface type. We ignore changes in displacement height as they are expected to be much less important than the changes in $z_o$ (Text S1). To derive the MODIS-Koeppen type land map (Steinkamp and Lawrence, 2011) required for soil $NO_x$ module, we first use translate the PFT map according to International Geosphere-Biosphere Programme (IGBP) land cover classification system (http://www.eomf.ou.edu/static/IGBP.pdf). We use global monthly temperature climatology (Matsuura

and Willmott, 2012) to further differentiate the land types by climate with criteria outlined by Kottek et al. (2006). Finally, the ESA CCI PFT map is converted to Community Land Model (CLM) PFT map, which is required for MEGAN BVOC emissions module, by the temperature criteria specified by Bonan et al. (2002). As the method of deriving C3:C4 grass ratio was subsequently updated (Lawrence and Chase, 2007), this ratio is directly taken from CLM land surface data set.

In the Supplementary Material, we provide an evaluation of the annual mean simulated SNA aerosol mass concentration and surface $O_3$ mixing ratios from the Simulation 4 (representative of 2014 conditions) with globally available observations from the same time period. In general, the model captures the spatial distributions of individual SNA species reasonably well (Fig. S1). The model is able to capture regional annual means of individual SNA species (Table S1) over Europe. Over the US and China, where annual means of all SNA species are underestimated by 21 – 55%, and in regions covered by Acid Deposition

Monitoring Network in East Asia (Japan, Korea and southeast Asia) where $SO_4^{2-}$ is underestimated by 36%, we expect the model may underestimate the sensitivity of SNA concentration to $NH_3$ emission perturbations. This may imply that results from our study should be interpreted as conservative. Figure S2 shows the reasonable agreement on annual mean surface $O_3$ between our model output and the gridded observation dataset from Sofen et al. (2016) (mean bias = +1.81 ppbv and mean absolute error = 3.97 ppbv). Our model therefore captures the present-day annual means of surface SNA and $O_3$ concentrations,

providing basis for our subsequent analyses. We also provide definitions for geographical regions, which largely follow Integrated modelling of global environmental change (IMAGE) 2.4 classifications, in Table S2.

**3 Changes in Land Cover, and Biospheric Fluxes, and Agricultural Emissions**

Table 2 shows the changes in global coverage of the major land cover types from 1992 to 2014 derived by the ESA CCI land cover product. The coverage of managed grass (including cropland and pasture) and built-up area, both of which are

unmistakably related to human activities, have increased mainly at the expense of forest coverage. This is consistent with a global trend in deforestation over this period. Figure 1 shows the spatial distribution of changes in fractional coverage of the major land cover types. Expansion of agricultural land at the expense of broadleaf forest coverage is most notable in South

America and Southeast Asia, which is well-documented in other studies based on remote sensing (Hansen et al., 2013) and national surveys (Keenan et al., 2015). The expansion of agricultural land over this time period is also observed in central Asia, southern China and Africa, but usually at the expense of land types other than broadleaf forests (mainly primary grassland and needleleaf forests). Meanwhile, transitions from agricultural land to forests and built-up areas is observed in northern China and eastern Europe, consistent with the findings of Potapov et al. (2015) and Lai et al. (2016).

Figure 2 shows the global changes in 3-year (2012-2014 minus 1991-1993) annual mean LAI calculated from the GLASS LAI data set. Over southern China and South America, the area with regionally consistent deforestation experience general increase in LAI, while the opposite effect is observed in Sahel and Former Soviet Union. In Europe, LAI increases in most parts despite a fairly consistent retraction of agricultural land is observed. The agricultural expansion and deforestation over Southeast Asia is mostly concurrent with the LAI decreases. LAI increases notably in northern China where agricultural land decreases. The fact that LAI change can be driven by factors other than changes in land cover type (e.g. temperature, precipitation, atmospheric $CO_2$ level) (e.g. Zhu et al., 2016) may explain the regionally divergent trends response of LAI to agricultural land use change. For example, the general increase of LAI in China is not only driven by changes in biome types, but also the greening within cropland (mainly attributable to agricultural intensification) and forests (mainly attributable to ambitious tree planting programmes) (Chen et al., 2019). Similarly, some deforested land in South America might have been cultivated intensively, resulting in an increase rather decrease in LAI. We also note that since the relationship between satellite-derived surface reflectance and retrieved LAI depends on land cover, the use of static land cover map in long-term LAI retrievals (Claverie et al., 2016; Xiao et al., 2016; Zhu et al., 2013) may not fully capture the effect of LULCC on LAI (Fang et al., 2013). In particular, Fang et al. (2013) show that LAI could be substantially overestimated when grasses and crops are misclassified as forest. We may therefore overestimate dry deposition velocity over regions with significant deforestation. Such impact on biogenic emissions is secondary as biogenic emissions are expected to be much more sensitive to land cover type than LAI (e.g. Guenther et al., 2012).

These changes in land cover produce changes in the biogenic fluxes of reactive trace gases between the Earth's surface and atmosphere derived by GEOS-Chem. Figure 3a shows the calculated changes in annual mean isoprene emission due to land cover change over 1992 to 2014, and suggests that global isoprene emission could have decreased by 5.12 Tg/yr (-1.5 %). The largest local reductions in isoprene emissions (up to 30%) are observed in parts of South America, where deforestation from highly isoprene-emitting broadleaf forests is most strongly observed. We note that the decrease of isoprene emission simulated in Southeast Asia does not agree with the result from Silva et al. (2016), since our remote sensing data does not have separate land cover class for oil palm plantations which have expanded dramatically in the region. Our model may not therefore capture the full effects of LULCC on isoprene emission, and its effect on $PM_{2.5}$ and $O_3$ over the region. Elsewhere in the world, the signals of land cover change on isoprene emissions are mostly small and follow the local patterns of changes in LAI. Changes in monoterpenes ($< 5$ ng m$^{-2}$ s$^{-1}$) and sesquiterpene ($< 1$ ng m$^{-2}$ s$^{-1}$) emissions are relatively small.

Figure 3b shows the changes in annual mean soil NO emission due to LULCC, which represent the change in soil emission driven purely by LAI (which affects canopy uptake) and land cover changes (which affects both biome-based emission factor and canopy uptake) (i.e. without considering the changes in nitrogen input). LULCC leads to a small signal of +0.04 TgN/yr (+0.6%) in global soil NO emission. The magnitude of changes in soil NO emission induced by LULCC is comparable to that in agricultural NO emissions inventory (see below) over certain regions (e.g. South America, Australia, Africa). Relatively large increases in soil NO is simulated over western Africa due to both cropland expansion and LAI reduction, which leads to smaller canopy reduction factor and larger emission factor.

Figure 3c shows the changes in annual mean $O_3$ dry deposition velocity ($v_d$), which also closely follow the pattern of LAI changes. Slight increases of $v_d$ are observed in China, India, Southeast US, central America, South America, Europe and southern Africa. In Southeast Asia $v_d$ decreases concurrently with deforestation and reduction in LAI. In central Brazil, the increase in LAI is offset by the deforestation of tropical evergreen broadleaf forests that have higher $v_d$ than other land types (Song-Miao Fan et al., 1990; Wang et al., 1998), leading to small overall change in $v_d$. Likewise, despite deforestation observed further south, these losses are offset by strong increases in LAI so that $v_d$ increases by up to 0.1 cm s$^{-1}$. Significant changes in the $v_d$ of $O_3$ due to LAI also imply that $v_d$ of other relevant trace gases (e.g. $NO_2$, $SO_2$) would also be perturbed by land cover change in our model, which will be discussed briefly in the subsequent section.

Figure 4 shows the changes in agricultural $NH_3$ and $NO_x$ emissions between 1992 and 2014, which consists mostly of emissions from fertilizer application and manure management (Hoesly et al., 2018). According to the CEDS inventory, global direct agricultural $NH_3$ emissions increased by 7.6 Tg N/yr since 1992, equivalent to a 19 % increase in total anthropogenic $NH_3$ emissions. Direct agricultural soil $NO_x$ emissions increased by 0.37 Tg N/yr since 1992, and while this is a substantial increase in agricultural soil $NO_x$ emissions (26 %), it represents only a 1 % increase in total anthropogenic $NO_x$ emissions.

The increases in agricultural emissions are most substantial over South Asia, followed by China, parts of Middle East, Southeast Asia and South America, and to a less degree in central America, North America and Sahel. The sharpest decline of agricultural emissions is observed in Europe and Former Soviet Union, followed by milder declines over Japan and Korea. The particularly sharp decline of agricultural emissions in Europe is mainly attributable to the implementation of emission control protocols (National Emissions Ceilings (NEC) and Integrated Pollution Prevention and Control (IPPC) directives) within the European Union (Skjøth and Hertel, 2013). According to the CEDS inventory, changes in agricultural emissions dominates the trend of total $NH_3$ emissions in all major regions except Africa, where large part of the $NH_3$ emissions trend is attributable to the waste management and the RCO (residential, commercial, other) sectors (Hoesly et al., 2018) (Figure S3). In contrast, the increase in agricultural emissions of $NO_x$ does not contribute significantly to the global increase of total $NO_x$ emissions over our period of concern.

We note that the hotspots of change in managed land cover and of change in agricultural emissions are not always overlapping. For example, agricultural emissions increase significantly over northern China and northern India, while the cropland coverage over those regions does not increase correspondingly over this same period. Such agricultural intensification in turn contribute significantly to the positive LAI trend over the above regions (Chen et al., 2019). Similarly, agricultural emissions have declined over Kazakhstan, while the area of managed land has not decreased significantly. This highlights a degree of independence between land management and LULCC, with both being components of land change but having potentially distinct spatial patterns and impacts on air quality. This also highlights the importance of treating both in our chemical transport model simulations as they occur contemporaneously around the globe, and may have different impacts on air quality.

## 4 Impact of LULCC and agricultural emission changes on surface PM$_{2.5}$

Figure 5 shows the modeled impacts of LULCC, changes in agricultural emissions, and the combined effects of both, on annual mean surface PM$_{2.5}$ (under 2014 anthropogenic emissions). We have calculated the impacts of LULCC on PM$_{2.5}$ ("$\Delta$PM$_{2.5,\ LULCC}$" as the difference in PM$_{2.5}$ predicted by Simulation 3 and Simulation 2; the impacts of agricultural emission changes on PM$_{2.5}$ ("$\Delta$PM$_{2.5,\ agr\_emis}$") as the difference in PM$_{2.5}$ predicted by Simulation 4 and Simulation 3; and the impacts of these combined ("$\Delta$PM$_{2.5,\ LULCC+agr\_emis}$") as the difference in PM$_{2.5}$ predicted Simulation 4 and Simulation 2 (see Table 1).

The effect of LULCC on PM$_{2.5}$ (Fig. 5a) is mainly through perturbing BVOC emissions as they are a precursor to SOA. Over parts of South America and Southeast Asia, where isoprene emissions drop significantly due to deforestation, PM$_{2.5}$ is reduced by up to 0.7 $\mu$g m$^{-3}$. Land cover changes also lead to changes in the dry deposition velocity of some SNA precursor gases where stomatal uptake is an important deposition pathway (e.g. NO$_2$ and SO$_2$, Fig. S4). Indeed, over India and China, where our model suggests high levels of SNA aerosol precursors, contemporary LULCC enhances dry deposition of these constituents which reduces PM$_{2.5}$ overall by up to 0.3 $\mu$g m$^{-3}$, similar to the finding of Fu et al. (2016).

We find that the agricultural emissions generally have larger impact on annual mean surface PM$_{2.5}$ level (Fig. 5b) than LULCC. The largest increases in annual mean surface PM$_{2.5}$ due to changes in agricultural emissions over 1992 to 2014 occur across China (+0.7 $\mu$g m$^{-3}$) and India (+1.6 $\mu$g m$^{-3}$). Over some hot spots in the two countries (e.g. northwestern India and North China Plain), the local changes in PM$_{2.5}$ exceed 3.5 $\mu$g m$^{-3}$, supporting the previously emphasized importance of controlling NH$_3$ emissions on PM air quality of China (Fu et al., 2017), but potentially India as well. Some moderate increases (< 2 $\mu$g m$^{-3}$ in most locations) in annual mean PM$_{2.5}$ concentrations are also observed in the Middle East, North America, central America and South America.

The largest decreases (up to 2.1 μg m$^{-3}$) in annual mean PM$_{2.5}$ due to changes in agricultural emissions are simulated in central and eastern Europe and Former Soviet Union. Despite comparable reductions in agricultural NH$_3$ emissions, decreases in PM$_{2.5}$ over western Europe are smaller because of weaker sensitivity of SNA aerosol to NH$_3$ emissions, which is consistent with the finding of Lee et al. (2015) and Pozzer et al. (2017). In general, reductions in annual mean PM$_{2.5}$ due to agricultural emission changes simulated over western Europe are weaker than over central and eastern Europe and Former Soviet Union.

Figure 5c shows the combined effect of agricultural emissions and LULCC on annual mean surface PM$_{2.5}$, which we have already shown is mostly dominated by the effect of agricultural emissions. Nevertheless, we find that the effects of LULCC are able to partially offset the increase in PM$_{2.5}$ due to agricultural emissions changes over China and India. These offsets are occurring in densely populated areas, so that the effects on population-weighted average (method described in text S2) PM$_{2.5}$ concentrations (see below), and therefore potentially exposure, may be noteworthy. This is discussed in further detail below.

We note that the difference between Figure S5a and Figure 5c illustrates how ΔPM$_{2.5, \text{ LULCC+agr\_emis}}$ is sensitive to the anthropogenic emissions background. We find that surface PM$_{2.5}$ over US, Europe and Former Soviet Union are less sensitive to NH$_3$ emissions under 2014 anthropogenic emissions background, since both SO$_2$ and NO$_x$ emissions in these regions have decreased significantly (> 42% for NO$_x$ and > 58% for SO$_2$) over 1992 and 2014. The opposite is simulated in over China and India, where SO$_2$ and NO$_x$ emissions have increased by >50%.

Table 3 summarizes the simulated effects of LULCC and agricultural emission changes on PM$_{2.5}$ air quality, and compares their magnitudes with the concomitant effects from direct anthropogenic emission changes ("ΔPM$_{2.5, \text{ anth}}$") over the same time period. We additionally compare area-averaged and population-weighted global and regional metrics. While the resolution of our simulations does not capture urban-scale gradients and non-linearities in urban chemistry, the use of population weighting allows us to explore whether signals of change in land cover or land management are concentrated over areas of high population, or whether they are primarily observed over less populated areas.

Globally, our model results estimate that the global population-weighted change in PM$_{2.5}$ resulting from LULCC and agricultural emission changes (+0.70 μg m$^{-3}$) is on the order of ~10% of the change in PM$_{2.5}$ resulting from direct anthropogenic emissions (+7.99 μg m$^{-3}$) over 1992 to 2014. Regionally, the largest impact of land change (ΔPM$_{2.5, \text{ LULCC+agr\_emis}}$) on population-weighted annual mean surface PM$_{2.5}$ is simulated over central and eastern Europe (-1.01 μg m$^{-3}$), Former Soviet Union (-1.00 μg m$^{-3}$), South Asia (+1.71 μg m$^{-3}$) and China (+1.45 μg m$^{-3}$). In most regions, the difference between population-weighted ΔPM$_{2.5, \text{ agr\_emis}}$ and ΔPM$_{2.5, \text{ LULCC+agr\_emis}}$ is very small (< ~0.05 μg m$^{-3}$) except in China (0.12 μg m$^{-3}$). Generally, the impacts of land change on population weighted ΔPM$_{2.5}$ have the same sign as the impacts of direct anthropogenic emissions. The only exception to this occurs over North America where anthropogenic NO$_x$ and SO$_2$ emissions have declined, but agricultural emissions have increased. This suggests that the increase in agricultural emissions over NAm has partially canceled

out the effects of other emission controls on PM$_{2.5}$, though this effect is small so far (~5%). In other regions, population-weighted $\Delta$PM$_{2.5,\ LULCC+agr\_emis}$ is generally on the order of 5% to 12% of changes due to direct anthropogenic emissions (e.g. in central and eastern Europe and western Europe). Notably, over Former Soviet Union, the Middle East and central America, $\Delta$PM$_{2.5,\ LULCC+agr\_emis}$ are much more comparable to the effect of anthropogenic emission changes (24%, 42%, and 208% respectively).

Our result shows that the impact of LULCC and land management changes on PM$_{2.5}$ is mainly from the agricultural emission changes, while LULCC can result in additional impacts in regions with high SNA precursor emissions (e.g. India, China) through modulating dry deposition. The magnitude of population-weighted $\Delta$PM$_{2.5,\ LULCC+agr\_emis}$ suggests that land change may contribute significantly to regional and global changes in human PM$_{2.5}$ exposures and that the effects of these changes are not isolated to low population regions. Particularly, over the regions experiencing rapid change in land use intensity (e.g. Former Soviet Union) or slow change in anthropogenic emissions (e.g. central America, the Middle East), the effects of land changes on particulate air pollution could be comparable (24% to 208%) to the effects of direct anthropogenic emission changes.

## 5 Impact on surface O$_3$

Figure 6 shows the modeled impacts of LULCC, changes in agricultural emissions, and the combined effects of both, on annual mean surface O$_3$ (under 2014 anthropogenic emissions). These changes are calculated identically as for PM$_{2.5}$ above: the impacts of LULCC on O$_3$ ("$\Delta$O$_{3,\ LULCC}$") is the difference in O$_3$ predicted by Simulation 3 and Simulation 2; the impacts of agricultural emission changes on O$_3$ ("$\Delta$O$_{3,\ agr\_emis}$") is the difference in PM$_{2.5}$ predicted by Simulation 4 and Simulation 3; and the impacts of these combined ("$\Delta$O$_{3,\ LULCC+agr\_emis}$") is the difference in PM$_{2.5}$ predicted Simulation 4 and Simulation 2 (see Table 1). We also use predictions of surface HNO$_3$/H$_2$O$_2$ ratios (Figure S6) as a proxy of VOC- vs NO$_x$- sensitive chemical O$_3$ production (Peng et al., 2006; Sillman, 1995) in our discussion of the results.

The modelled response of surface O$_3$ to LULCC ($\Delta$O$_{3,\ LULCC}$) (Fig. 6a) involves several distinct processes (dry deposition, soil NO$_x$, and BVOC emissions). Over parts of North America and central America, the increase in dry deposition velocity ($v_d$) reduces annual mean surface ozone by up to 0.5 ppbv overall. In central Brazil, deforestation of tropical rainforests leads to significant reduction is isoprene emissions, reducing surface ozone by up to 0.8 ppbv in this NO$_x$-limited environment (Fig. S6). In contrast, modelled surface ozone decreases by up to 1.2 ppbv further south, where strong increases in LAI lead to largely increases $v_d$ (up to 0.06 m s$^{-1}$). The modelled reduction of surface ozone (up to 1 ppbv) over central African rainforests is also likely attributable to increased $v_d$ as neither soil NO$_x$ nor isoprene emissions change much in the region. However, in other parts of Africa, up to 0.6 ppbv of surface ozone increases are simulated, mainly because of the relatively large increase in soil NO emission. In southern China, up to 0.5 ppbv reduction in surface ozone is simulated, which is likely attributable to

the increase in $v_d$, and slightly offset by the small increase in isoprene emission under this $NO_x$-saturated environment (Fig. S6). Small surface $O_3$ changes, mainly due to transport, are also simulated over the Atlantic Ocean.

Overall, the role of agricultural emission changes in fertilizer-associated $NO_x$ plays a minor role in surface $O_3$ changes (Fig. 6b). An exception to this is observed in the large increase in agricultural $NO_x$ emissions which reduce surface $O_3$ by up to 0.6 ppbv over $NO_x$-saturated India and China, but increase surface $O_3$ in $NO_x$-limited parts of Southeast Asia by similar magnitude. Slight increases in surface $O_3$ level due to increased agricultural $NO_x$ emissions are also simulated over parts of eastern Africa and South America. Whether the effect of agricultural emissions strengthens (e.g. China and Sahel) or offsets (e.g. over

southern Brazil and India) the effect of LULCC is largely region-dependent. As shown in Fig. 6c, LULCC tend to dominate the impacts on surface $O_3$ over most regions in the world (unlike $PM_{2.5}$ where the effects of agricultural emission changes dominate).

Similar to $PM_{2.5}$, we find that the changes in anthropogenic emission background over 1992 to 2014 is strong enough to alter

the sensitivity of $O_3$ to land change. As indicated by Fig. S6, Asia was less $NO_x$-saturated, while western Europe and coastal United States were more $NO_x$-saturated in 1992 than in 2014. For example, the increase in soil NO emission over India is more likely to increase rather than decrease surface ozone concentration (fig. S7a), leading to different modelled effect on surface ozone.

Table 4 shows the change in area and population-weighted annual mean afternoon surface $O_3$ due to the effects of anthropogenic emissions ("$\Delta O_{3,\ anth}$", Fig. S7b), $\Delta O_{3,\ LULCC}$, $\Delta O_{3,\ agr\_emis}$ and $\Delta O_{3,\ LULCC+agr\_emis}$. In most regions, $\Delta O_{3,\ LULCC+agr\_emis}$ is positive. However, this is offset by the negative population-weighted average $\Delta O_{3,\ LULCC+agr\_emis}$ over the most populous regions (South Asia and China), resulting in very small globally averaged population-weighted $\Delta O_{3,\ LULCC+agr\_emis}$.

The magnitudes of population-weighted $\Delta O_3$ (within ±0.5 ppbv) display less regional variability than that of $\Delta PM_{2.5}$. Over Eastern Africa, Western Africa and Southern Africa, area-averaged $\Delta O_{3,\ LULCC+agr\_emis}$ generally has similar magnitudes to population-weighted $\Delta O_{3,\ LULCC+agr\_emis}$. In other regions, the differences between area and population-weighted $\Delta O_{3,\ LULCC+agr\_emis}$ are more substantial. The largest discrepancies between area and population-weighted $\Delta O_{3,\ LULCC+agr\_emis}$ is found over China, where increases in surface $O_3$ are predicted over less populated western China, while reductions in surface $O_3$ are

simulated over more densely-populated eastern China. In South America, there are large sub-regional signals of $\Delta O_{3,\ LULCC+agr\_emis}$, but these positive and negative largely offset each other, resulting both in small area-weighted and population-weighted $\Delta O_{3,\ LULCC+agr\_emis}$.

Over China, western Africa, eastern Africa, southern Africa, Former Soviet Union and the Middle East, the magnitudes of

population-weighted $\Delta O_{3,\ LULCC+agr\_emis}$ are more than 20% of that of $\Delta O_{3,\ anth}$, implying that contemporary land system changes

could be a regionally important component in contemporary trends of surface $O_3$. The effects of agricultural emission changes and LULCC can either noticeably enhance (e.g. over the Middle East, Japan and Korea, China) or offset (e.g. over South Asia) each other because of the dependence of $\Delta O_{3,\ agr\_emis+land\_cover}$ on regional $NO_x$-VOC chemistry and details of LULCC, indicating the complexity of diagnosing the effect of land change on surface $O_3$ at regional and global scale.


Our result suggests that contemporary agricultural emission changes and LULCC each have distinct effects on surface $O_3$, with LULCC generally stronger in magnitude. Both of the effects are dependent on local $NO_x$-VOC chemistry, as agricultural emission changes perturb $NO_x$ emissions, while LULCC tends to affect BVOC emissions. In addition, LULCC is also able to affect surface $O_3$ (and other precursors) directly through dry deposition through LAI changes over our period of concern. These effects are found to affect $O_3$ pollution over densely populated regions (e.g. China) and could be comparable to the magnitudes of $O_3$ changes due to anthropogenic emissions over specific regions (e.g. Former Soviet Union, eastern Africa, western Africa), indicating the importance of land change in studying long-term changes in surface $O_3$.

## 6 Impact on Nitrogen Deposition

Finally, we estimate the effect of these land changes on nitrogen deposition estimates. Figure 7 shows the global impact of LULCC and agricultural emission changes on total nitrogen deposition ($\Delta N_{dep}$), and Table 5 summarizes the regional and global results. The largest increase and decrease in nitrogen deposition ($N_{dep}$) are simulated over South Asia (+1.91 TgN yr$^{-1}$) and Former Soviet Union (-1.28 TgN yr$^{-1}$), respectively. Notable increases in $N_{dep}$ are also simulated over China (+1.55 TgN yr$^{-1}$), South America (+1.24 TgN yr$^{-1}$), North America (+0.66 TgN yr$^{-1}$), western Africa (+0.39 TgN yr$^{-1}$) and eastern Africa (+0.41 TgN yr$^{-1}$). Figure 7 also illustrates the simulated changes over 1992 to 2014 in area with nitrogen deposition ($N_{dep}$) exceeding 5 kgN ha$^{-1}$ yr$^{-1}$, which is a proxy of possible exceedance of critical $N_{dep}$ loads for terrestrial and fresh water (Moriarty, 1988).

Globally, there is a net increase in land area with $N_{dep}$ > 5 kgN ha$^{-1}$ yr$^{-1}$ of $3.67\times10^6$ km$^2$. The increase is mostly simulated over the Americas, Africa, the Middle East and China, which is partially offset the large decrease over Former Soviet Union. Meanwhile, despite agricultural changes that lead to notable $\Delta N_{dep}$, over most of Europe, eastern US, China, South Asia and Southeast Asia, nitrogen input from other sources are large enough that this signal alone does not lead to substantial changes in $N_{dep}$ exceedances of 5 kgN ha$^{-1}$ yr$^{-1}$. However, over parts of North America, South America, Africa and China, agricultural changes are simulated to increase $N_{dep}$ from below to above 5 kgN ha$^{-1}$ yr$^{-1}$. This implies these natural ecosystems at the edge of these areas are at risk of nitrogen exceedances due to agricultural changes. In contrast, the substantial reduction of $N_{dep}$ in parts of Former Soviet Union may have significantly reduce the risk of nitrogen exceedance of natural ecosystem from agricultural sources.

## 7 Discussion and Conclusions

In this work, we have explored how changes in the global land system, through LULCC and agricultural emission changes, may have impacted contemporary global air quality over 1992 to 2014. We model the effects of contemporary LULCC and
agricultural emission changes, individually then in combination, on surface $O_3$ and $PM_{2.5}$ using the GEOS-Chem CTM. With a uniquely consistent framework, we are able to integrate direct information from global emission inventories (CEDS) with updated land surface remote sensing products (ESA CCI land cover and GLASS LAI). This allows us to avoid invoking extra assumptions on land management practices (e.g. constant $N_r$ input, emissions or emission factors over time) and biophysical properties of PFTs (e.g. constant PFT-specific LAI over time).


We find that changes in agricultural emissions are simulated to increase the annual mean surface $PM_{2.5}$ concentrations in China and India by up to 3.5 µg m$^{-3}$ and to decrease in Europe by up to 3.5 µg m$^{-3}$. Our simulation suggests that though $\Delta PM_{2.5}$ is mainly attributable to changes in agricultural emissions in global scale, LULCC over India and China can lead to enhanced dry deposition of certain $PM_{2.5}$ precursor gases ($SO_2$ and $NO_2$), thus partially offsetting (~10 %) the increase in $PM_{2.5}$ from
agricultural regions. This implies a potentially important role of LULCC in determining SNA aerosol level over certain heavily polluted regions. Also, LULCC reduces BVOC emissions over Amazonia which lead to reductions in $PM_{2.5}$ by up to 0.7 µg m$^{-3}$. In a future with decreasing anthropogenic $NO_x$ and $SO_2$ emissions, which could diminish the importance of agricultural emissions on $PM_{2.5}$ formation (Bauer et al., 2016), LULCC may become increasingly important in the overall effect of land change on $PM_{2.5}$. Noticeable changes (> 1 µg m$^{-3}$) population-weighted $\Delta PM_{2.5, LULCC+agr\_emis}$ are simulated over China (+1.45
µg m$^{-3}$), South Asia (+1.71 µg m$^{-3}$), central and eastern Europe (-1.00 µg m$^{-3}$) and Former Soviet Union (-1.01 µg m$^{-3}$), indicating the potential impact of land change on long-term public health through modulating $PM_{2.5}$ level at regional scale. Our results suggest that contemporary (1996-2014) LULCC and agricultural emission changes contribute to changes in $PM_{2.5}$ at regional and global scales that range from on the order of 5 to 10% of changes in $PM_{2.5}$ resulting from direct anthropogenic emissions over the same time period, and up to ~25% or more in Former Soviet Union, the Middle East specifically.


In contrast, the effect of LULCC is generally stronger than that of agricultural emission change in simulations of surface $O_3$. We find that the role of LULCC over 1992 to 2014 is regionally significant enough to induce changes in BVOC emissions and dry deposition which affect surface $O_3$, but that the overall effects largely offset each other on the global scale, leading to very small population-weighted $\Delta O_{3, LULCC+agr\_emis}$. This finding with consistent with that of Ganzeveld et al. (2010), even the
timeframe of study (2000 – 2050) is different. Both the effects of agricultural emission changes and LULCC, through $NO_x$ and BVOC emissions, are sensitive to regional ozone production regime. The increase in agricultural emissions reduces $O_3$ over $NO_x$-saturated parts of China and South Asia by up to 0.6 ppbv, while the reduction in BVOC emissions increases surface $O_3$ over VOC-limited Amazonia by up to 1.2 ppbv; enhancements of dry deposition reduce $O_3$ over parts of China, North America and South America by up to 1.2 ppbv. Overall, the largest population-weighted $\Delta O_{3, LULCC+agr\_emis}$ is simulated over western

Africa (+0.42 ppbv) and eastern Africa (+0.47 ppbv). We find that the ratio between $\Delta O_{3,\ LULCC+agr\_emis}$ and $\Delta O_{3,\ anth}$ varies widely depending on region, with some having $\Delta O_{3,\ LULCC+agr\_emis}$ that are comparable (>20%) to $\Delta O_{3,\ anth}$. These results show the complexity and importance of land change on mediating long-term changes in surface $O_3$.

    We also find that both the modelled $\Delta O_{3,\ LULCC+agr\_emis}$ and $\Delta PM_{2.5,\ LULCC+agr\_emis}$ are sensitive to the changes in anthropogenic

emissions suggested by CEDS inventory over 1992 and 2014, as the changes in $NO_x$, $SO_2$ and VOC emissions are large enough to perturb atmospheric $HNO_3$ and $H_2SO_4$ production, and ozone production regime considerably in many regions (e.g. Asia and western Europe). This highlights the necessity of accurate and relevant emission inventories when evaluating the impacts of land change on air quality (e.g. Bauer et al., 2016).

The increased atmospheric reactive nitrogen (+7.20 Tg/yr) due to agricultural emissions is mostly found to deposit near to source region as the atmospheric lifetime of $NH_3$ is generally short, which implies the potential risk of excessive nitrogen input over the natural ecosystems near to the regions with increases in agricultural emissions.

    Our work suggests that, at contemporary timescales (on the order of ~20 years), the effect of land change on air quality can

sometimes be important relative to the air quality changes induced by trends in direct anthropogenic emissions. We also find that agricultural emission changes have stronger effects on $PM_{2.5}$, while LULCC have stronger effects on $O_3$. This finding is comparable to that from Heald and Geddes (2016), which suggest a much more comparable changes in biogenic SOA (mostly induced by LULCC) and particulate nitrate (mostly induced by agricultural emission changes), and stronger surface ozone changes induced by land change over 1850 – 2000. This shows that both the magnitudes and relative contributions from

different components of land change effects on air quality vary significantly as timescale of study, and its potential importance at longer timescales (e.g. multidecadal, centennial), despite the relative small signal that we obtain here.

    We find the effects of agricultural emissions and LULCC to be largely linearly additive over contemporary timescales, which may be attributable to two factors: 1) LULCC mainly impacts $O_3$ precursors while agricultural emissions mainly impact SNA

precursors, and these are often spatially segregated; 2) LULCC and agriculture-related changes in surface fluxes of $O_3$ and SNA precursors are not large enough to change their respective chemical production regime. At longer timescale when land change signals are stronger, the effects of LULCC and agricultural emissions may be non-linear.

    We note several important limitations and opportunities for development. We were only able to evaluate our simulation

extensively over Europe, North American and East Asia. In most other regions where such evaluation of SNA speciation is not feasible, the sensitivity of SNA formation to $NH_3$ emissions can be a major source of uncertainty. Given the changes in agricultural emissions have occurred in global scale, effort of monitoring SNA speciation outside North America and Europe (e.g. Weagle et al., 2018) is necessary for understanding the sensitivity of $PM_{2.5}$ to agricultural emissions in global extent.

Better understanding of both the sources and sinks of $HNO_3$ (e.g. Heald et al., 2012; Holmes et al., 2019; Luo et al., 2019; Petetin et al., 2016) and nitrate partitioning (e.g. Vasilakos et al., 2018) are important for modelling SNA aerosol and its sensitivity to $NH_3$ emissions. Agricultural $NO_x$ and $NH_3$ emissions estimates also carry large uncertainty due their biological nature and resulting dependence on environmental conditions, which are not explicitly considered in the construction of bottom-up anthropogenic emission inventories (Crippa et al., 2018; Hoesly et al., 2018). Bidirectional exchanges of $NO_2$ (Breuninger et al., 2013; Chaparro-Suarez et al., 2011; Lerdau et al., 2000) and $NH_3$ (Bash et al., 2013; Massad et al., 2010; Wichink Kruit et al., 2012; Zhang et al., 2010)are not explicitly modelled (although in some regions may be implicitly accounted for in the regional scaling performed by CEDS), which introduces some uncertainty in the accuracy of surface flux modelling. Zhu et al. (2015) implemented a bi-directional $NH_3$ exchange model in GEOS-Chem, and found no substantial improvement with observations in the modelled $NH_3$ concentration, $NH_4^+$ wet deposition and nitrate aerosol concentration compared to the default GEOS-Chem uni-directional exchange framework. This indicates the uni-directional framework may still be sufficiently accurate in simulating global air quality comparing to bi-directional framework, which requires more observations to properly parameterize at global scale. In the case of $NO_2$, we make the assumption that in most regions we are interested in (fig. S9), the ambient concentrations of $NO_2$ exceed an ecosystem compensation point (0.05-0.6 ppb) (e.g. Breuninger et al. 2013) so that we can assume deposition would dominate. The simplistic representation of dry deposition in general, particularly the lack of dependence of stomatal conductance on atmospheric and soil water content, may not adequately capture the effects of LULCC, as biomes can have differential responses to meteorological and hydrological conditions. The inherent inconsistency of long-term LAI time series derived from reflectance measured by different instruments (Jiang et al., 2017) and the use of static land cover maps also introduce uncertainty in the LAI retrieval (Fang et al., 2013) and the subsequently computed LAI changes and trends, and these have been shown to be important to changes in simulated $O_3$ in this study and elsewhere (Wong et al. 2019). Though the use of PFT-based emission factors in regional and global modelling is generally justifiable (Guenther et al., 2012), we cannot rule out the possibility of intra-PFT variabilities of BVOC emission factors affecting the accuracies our results, which is exemplified by the inability of our model to capture the palm-driven isoprene emission increase over Southeast Asia (Silva et al., 2016) as discussed in section 3. Finally, the meteorological feedbacks (e.g. changes in sensible heat, latent heat, air temperature, boundary layer height) and the subsequent effects on atmospheric chemistry and transport from LULCC and agricultural emissions are not considered in our study, which could potentially be important (e.g. Wang et al., 2020).

Our study helps demonstrate the possible magnitudes and regional patterns of the impacts of contemporary LULCC and agricultural emission changes on $PM_{2.5}$ and $O_3$, and suggests that the combination of these factors should not be neglected in the study of regional and global air quality changes over multi-decade timescales. Our results confirm the potential importance of controlling agricultural emissions on improving $PM_{2.5}$ air quality, which could be practical as numerous feasible options exist for reducing agricultural emissions through optimizing livestock and crop production system (e.g. Ti et al., 2019). Incentivizing these and other practices that improve agricultural nitrogen use efficiency (e.g. including livestock production

with cropping, synchronizing nitrogen supply with crop demand) (e.g. Fageria and Baligar, 2005; Langholtz et al., 2021) can be one of the keys to mitigate the air quality impacts of reactive nitrogen input without compromising agricultural productivity

(e.g. Guo et al., 2020). Furthermore, as increasing reactive nitrogen input and land use change are the two of the main strategies to meet the global demand for biomass-based products in the future (Foley et al., 2011), the distinct yet significant impacts of agricultural emissions and land use change on $O_3$, $PM_{2.5}$ and nitrogen deposition should be investigated as part of the overall environmental impacts of land system changes, especially when tradeoff between increasing land input and cropland expansion exists (e.g. Lotze-Campen et al., 2010; Mauser et al., 2015). This could benefit agricultural policy activities by appropriately

considering all in the externalities and socioeconomic costs of different options and scenarios for agricultural expansion.

**Code Availability**

The source code of GEOS-Chem model is publically available (https://doi.org/10.5281/zenodo.3634864). The GEOS-Chem model output and other source code used in the project can be obtained by contacting the corresponding author (jgeddes@bu.edu).

**Competing interests**

The authors declare that they have no conflict of interest.

**Author Contributions**

AYHW and JAG developed the ideas for this study, formulated the methods, and designed the model experiments together. AYHW performed the chemical transport model simulations and data analysis, with input and feedback from JAG. Manuscript

preparation was performed by AYHW, reviewed, edited, and approved by JAG.

**Acknowledgement**

This work was funded by an NSF CAREER grant (ATM-1750328) to project PI J.A. Geddes. We also thank the Global Modelling and Assimilation Office (GMAO) at NASA Goddard Flight Center for providing the MERRA-2 data, European Space Agency Climate Change Initiative (ESA CCI) for the land cover time series, Center for Global Change Data Processing

and Analysis at Beijing Normal University (BNU) for GLASS LAI product.

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

| | 1 | 2 | 3 | 4 | 5 |
|---|---|---|---|---|---|
| **Agricultural emissions** | 1992 | 1992 | 1992 | 2014 | 2014 |
| **Land cover** | 1992 | 1992 | 2014 | 2014 | 2014 |
| **Anthropogenic emissions** | 1992 | 2014 | 2014 | 2014 | 1992 |

**Table 1. Model configurations. The numbers in the top row are referred to in the main text.**


| Land cover | Coverage at 1992 (km$^2$) | Coverage at 2015 (km$^2$) | Change (km$^2$) |
|---|---|---|---|
| **Needleleaf forest** | $1.115 \times 10^7$ | $1.106 \times 10^7$ | $-8.892 \times 10^4$ (-0.8%) |
| **Broadleaf forest** | $2.146 \times 10^7$ | $2.092 \times 10^7$ | $-5.409 \times 10^5$ (-2.5%) |
| **Natural grass and shrub** | $3.769 \times 10^7$ | $3.768 \times 10^7$ | $-2.067 \times 10^4$ (-0.1%) |
| **Managed grass** | $2.127 \times 10^7$ | $2.199 \times 10^7$ | $+7.157 \times 10^5$ (+3.4%) |
| **Built-up area** | $2.603 \times 10^5$ | $5.966 \times 10^5$ | $+2.948 \times 10^5$ (+113%) |

**Table 2. Global LULCC summarized by the changes in coverage of different land types (2014 – 1992) from ESACCI land cover product**



| Region[†] | $\Delta PM_{2.5(anth)}$ | $\Delta PM_{2.5(LULCC)}$ | $\Delta PM_{2.5(agr\_emis)}$ | $\Delta PM_{2.5(LULCC+agr\_emis)}$ |
|---|---|---|---|---|
| FSU | -1.33 (-4.18) | +0.00 (+0.01) | -0.42 (-1.02) | -0.41 (-1.00) |
| CEU | -7.36 (-8.14) | -0.01 (-0.01) | -0.90 (-0.99) | -0.90 (-0.99) |
| WEU | -4.01 (-8.40) | -0.01 (-0.01) | -0.19 (-0.41) | -0.20 (-0.42) |
| China | +8.32 (+19.6) | -0.03 (-0.11) | +0.72 (+1.57) | +0.70 (+1.45) |
| SAs | +11.6 (+17.6) | -0.02 (-0.05) | +1.21 (+1.77) | +1.19 (+1.71) |
| ME | +1.16 (+1.06) | +0.01 (+0.01) | +0.29 (+0.43) | +0.30 (+0.44) |
| NAm | -1.58 (-5.44) | -0.00 (-0.01) | +0.07 (+0.28) | +0.07 (+0.27) |
| CAm | -0.37 (-0.12) | -0.01 (-0.01) | +0.11 (+0.25) | +0.11 (+0.25) |
| Global | +0.22 (+7.99) | -0.01 (-0.04) | +0.01 (+0.74) | +0.00 (+0.70) |

**Table 3. Changes in area averaged, and population-weighted (in parentheses), annual mean surface PM$_{2.5}$ concentrations (in µg m$^{-3}$) due to anthropogenic emissions alone ($\Delta PM_{2.5(anth)}$), LULCC ($\Delta PM_{2.5(LULCC)}$), agricultural emissions ($\Delta PM_{2.5(agr\_emis)}$), and the combined effects of LULCC and agricultural emission ($\Delta PM_{2.5(LULCC+agr\_emis)}$) together. Results only from regions with $\Delta PM_{2.5(LULCC)}$, $\Delta PM_{2.5(agr\_emis)}$ or $\Delta PM_{2.5(LULCC+agr\_emis)} > 0.2$ µg m$^{-3}$ are shown. [†]The definitions and abbreviations of all regions can be found in Table S2.**

| Region[†] | $\Delta O_{3(anth)}$ | $\Delta O_{3(LULCC)}$ | $\Delta O_{3(agr\_emis)}$ | $\Delta O_{3(LULCC+agr\_emis)}$ |
|---|---|---|---|---|
| FSU | -0.06 (+0.41) | +0.08 (+0.25) | +0.02 (+0.00) | +0.10 (+0.25) |
| China | +1.41 (-1.13) | +0.20 (-0.10) | +0.00 (-0.14) | +0.21 (-0.24) |
| SAs | +3.80 (+3.41) | +0.35 (+0.25) | -0.12 (-0.25) | +0.22 (-0.01) |
| ME | +1.98 (+0.74) | +0.35 (+0.28) | -0.05 (-0.06) | +0.31 (+0.23) |
| WAf | +1.22 (+1.99) | +0.36 (+0.33) | +0.05 (+0.08) | +0.41 (+0.42) |
| SAf | +0.95 (+1.10) | +0.30 (+0.25) | +0.02 (+0.02) | +0.32 (+0.27) |
| EAf | +1.53 (+1.92) | +0.31 (+0.29) | +0.10 (+0.18) | +0.41 (+0.47) |
| Global | +0.79 (+1.70) | +0.09 (+0.08) | +0.02 (-0.06) | +0.11 (+0.02) |

**Table 4. Changes in area averaged and population-weighted (in parentheses) annual mean surface O$_3$ concentrations (in ppbv) due to anthropogenic emissions alone ($\Delta PM_{2.5(anth)}$), LULCC ($\Delta PM_{2.5(LULCC)}$), agricultural emissions ($\Delta PM_{2.5(agr\_emis)}$), and the combined effects of LULCC and agricultural emission ($\Delta PM_{2.5(LULCC+agr\_emis)}$) together. Results only from regions with population-weighted average $\Delta O_{3(LULCC)}$, $\Delta O_{3(agr\_emis)}$ or $\Delta O_{3(LULCC+agr\_emis)} > 0.2$ ppb are shown. [†]The definitions and abbreviations of all regions can be found in Table S2.**

| Region[†] | $\Delta N_{dep}$ (TgN yr$^{-1}$) | $\Delta Area_{crit}$ (1000 km$^2$) |
|---|---|---|
| FSU | -1.28 | -1064 |
| China | +1.55 | +502 |
| SAs | +1.91 | 0 |
| ME | +0.29 | +494 |
| SEA | +0.61 | +244 |
| NAm | +0.66 | +788 |
| SAm | +1.24 | +1467 |
| WAf | +0.39 | +487 |
| SAf | +0.15 | +363 |
| EAf | +0.41 | +364 |
| Global | +7.20 | +3673 |

**Table 5. Changes in total nitrogen deposition ($\Delta N_{dep}$) and land area that has nitrogen deposition > 5 kgN ha$^{-1}$ yr$^{-1}$ ($\Delta Area_{crit}$), which is a proxy of potential risk of critical nitrogen deposition load exceedance. Only regions with significant $\Delta N_{dep}$ (> 0.25 TgN yr$^{-1}$) or $\Delta Area_{crit}$ (> $10^5$ km$^2$) are shown.**

**[†]The definitions and abbreviations of all regions can be found in Table S2.**

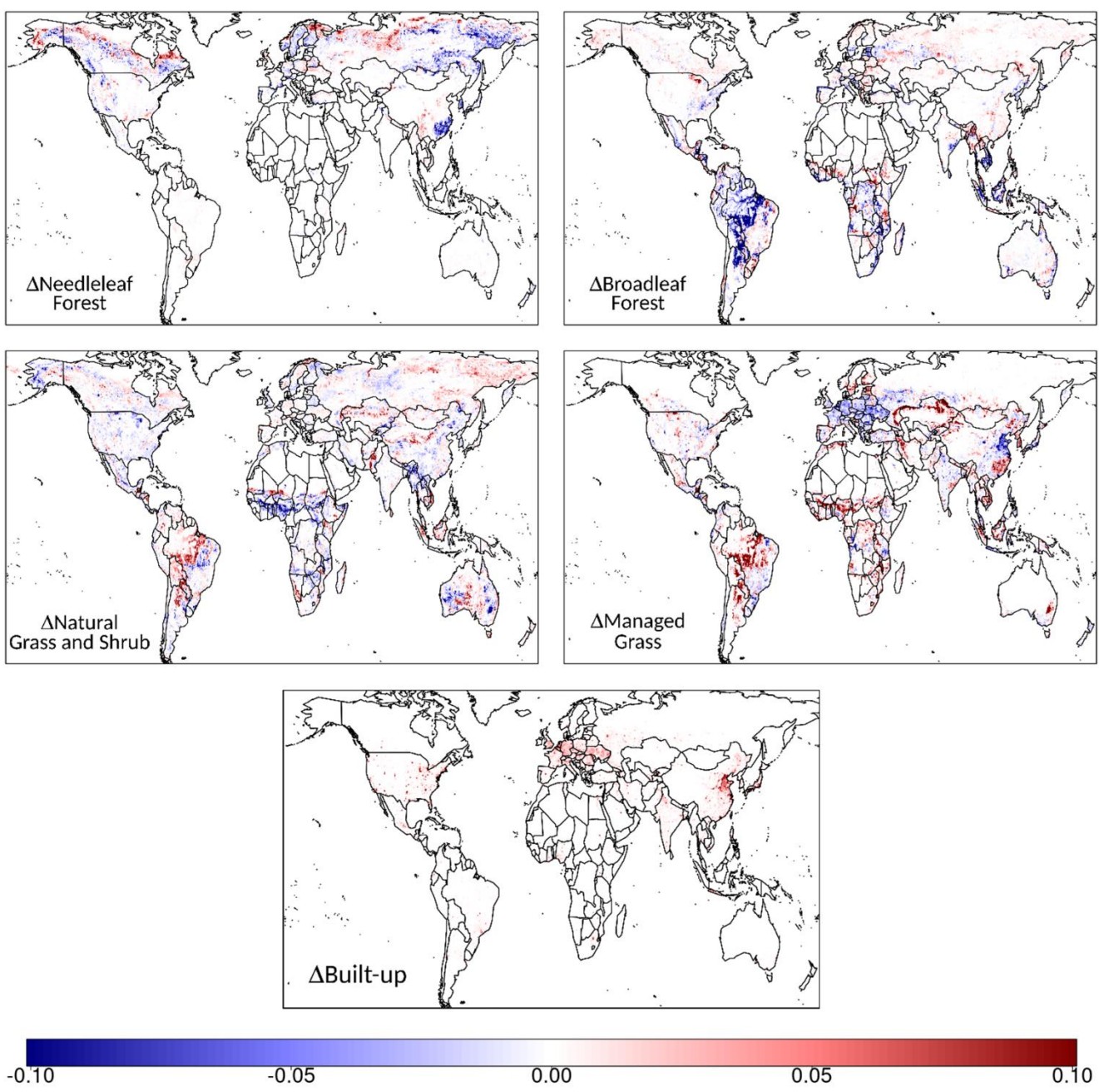

Figure 1. Global spatial patterns of 2014-1992 LULCC characterized by the changes in fractional coverages within a grid box (unitless) of major land cover types derived from the ESA CCI land cover product.

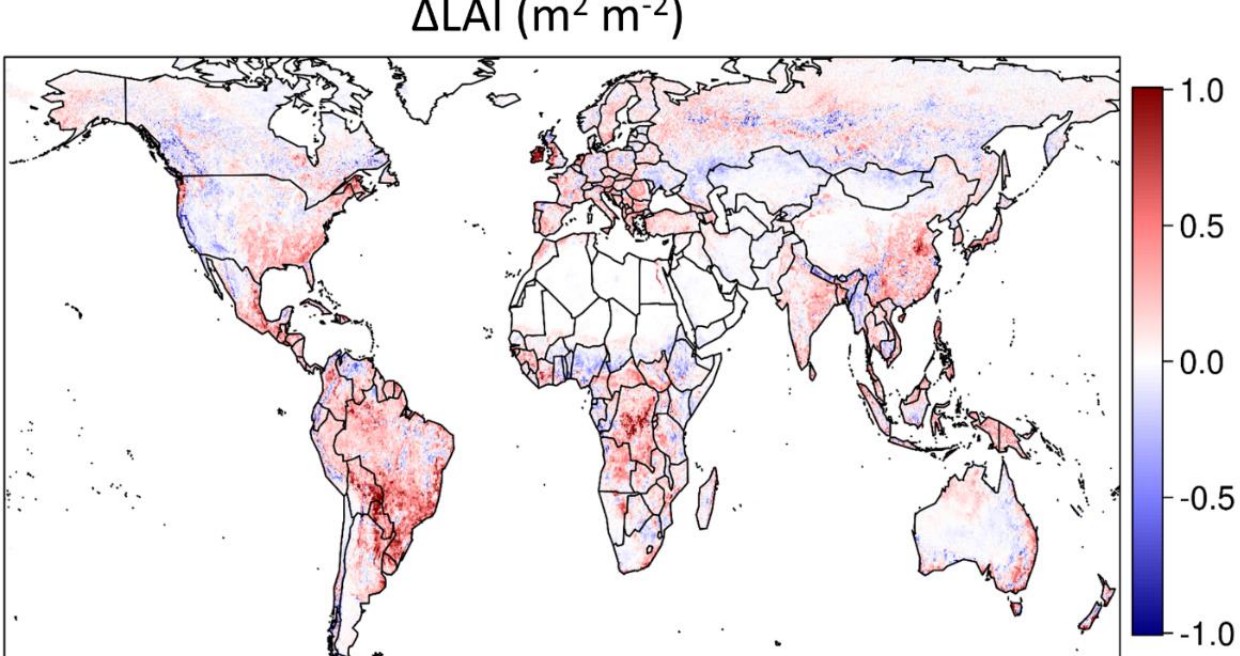

**Figure 2. Global Land Surface Satellite (GLASS)-derived changes in 3-year mean annual leaf area index (LAI) (2012 to 2014 average minus 1991 to 1993 average).**

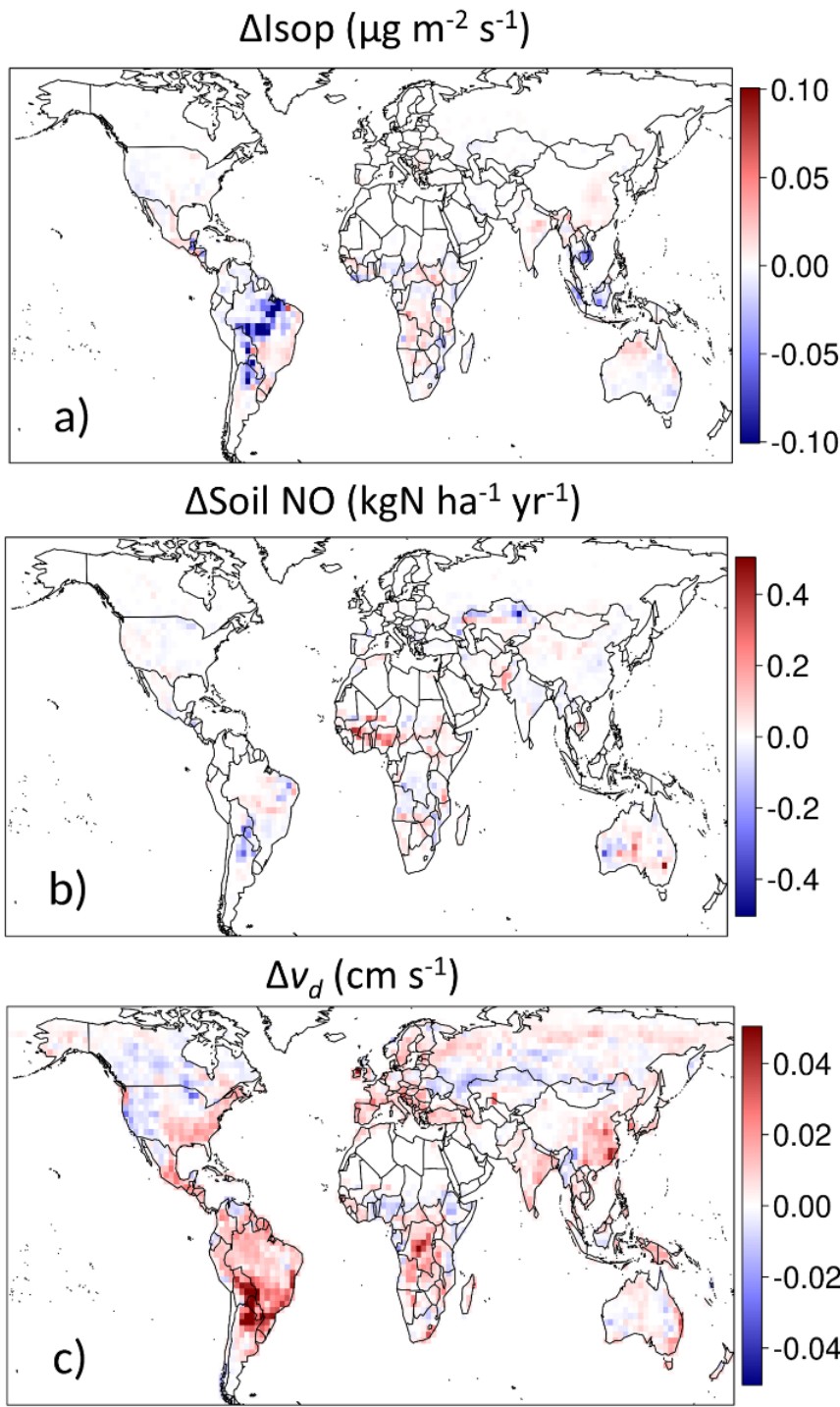

855

**Figure 3. Changes in annual mean a) isoprene emission (ΔIsop), b) soil NO emission (ΔSoil NO), and c) O₃ dry deposition velocity (Δ$v_d$), due to LULCC over 1992 and 2014.**

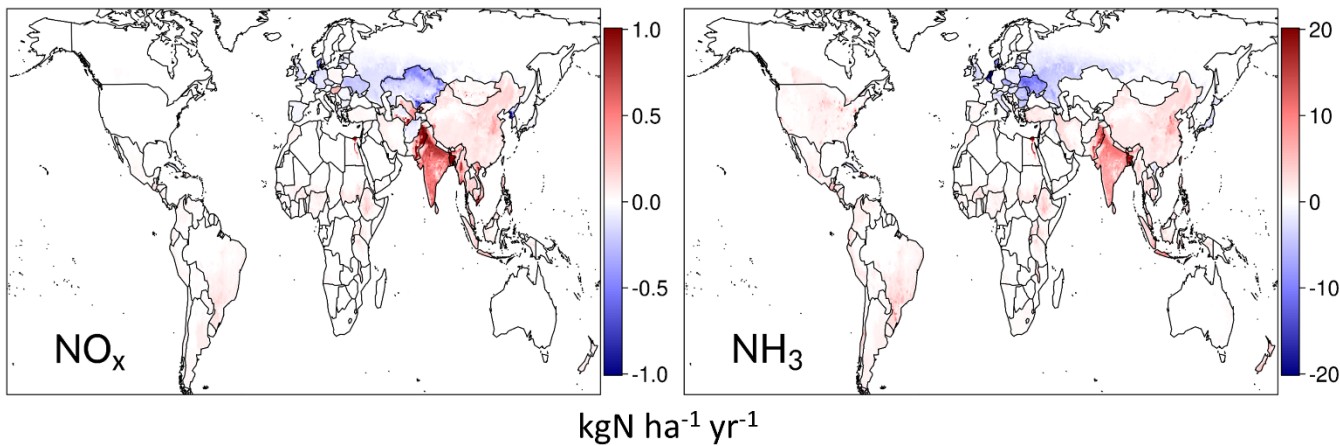

**Figure 4. Changes in agricultural NH₃ and NOₓ emissions (2014 – 1992) as implemented by the Community Emissions Data System (CEDS).**

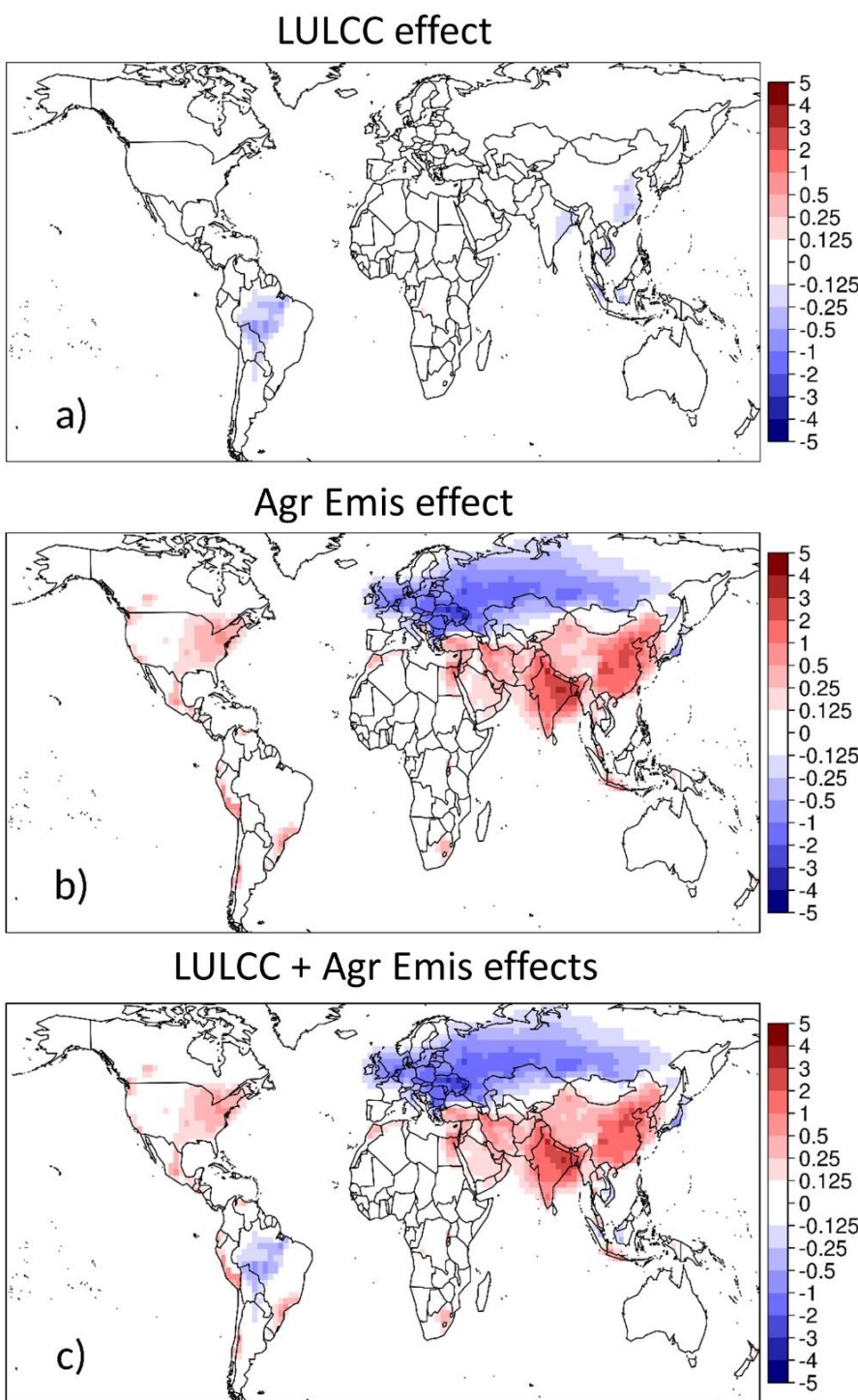

**Figure 5. Simulated changes in annual mean surface PM₂.₅ due to (a) LULCC, (b) agricultural emission ("Agr Emis") changes, and (c) the combined effects of agricultural emissions and LULCC.**

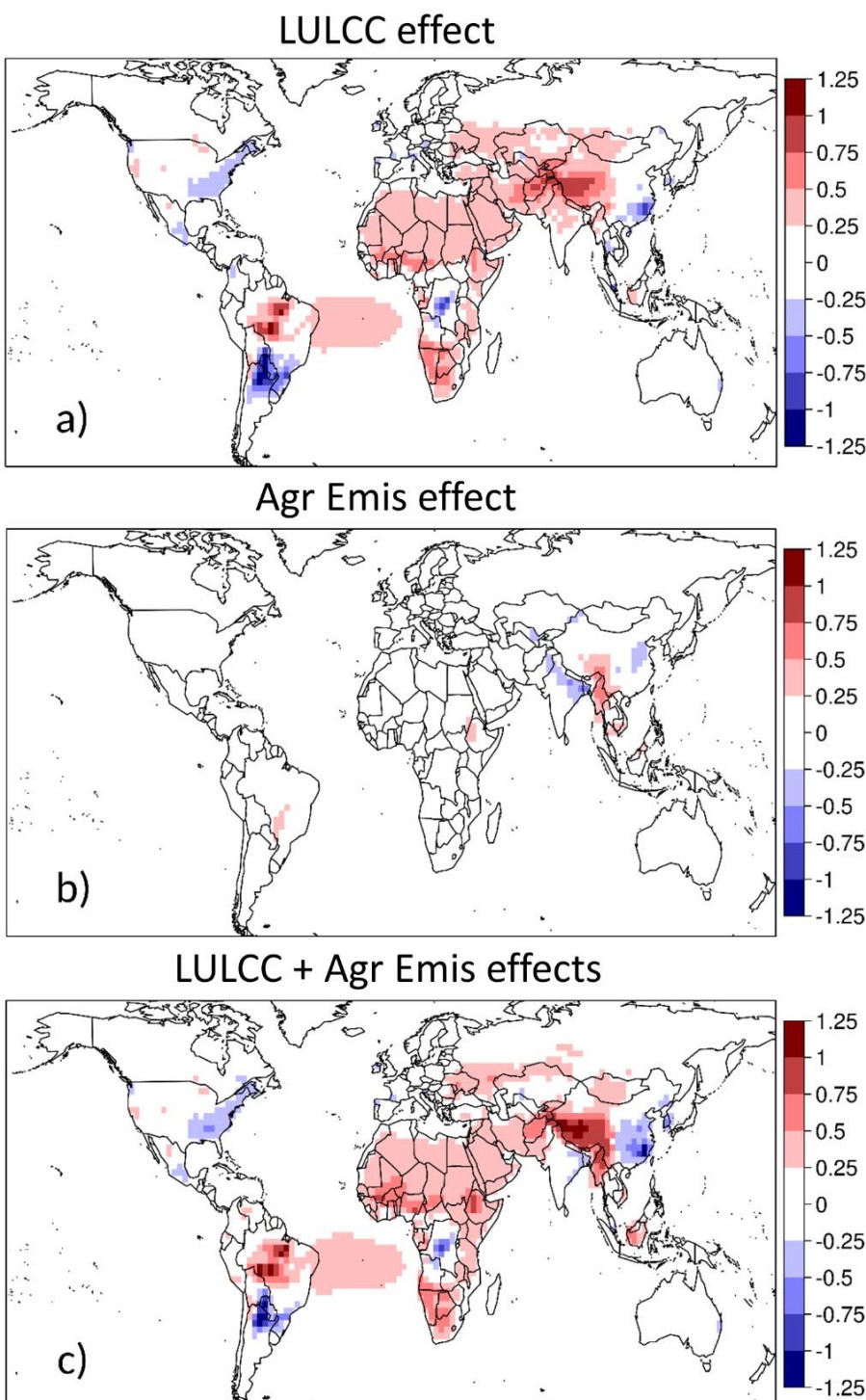

865  **Figure 6. Simulated changes in annual mean surface O₃ due to (a) LULCC, (b) agricultural emission ("Agr Emis")**
**changes, and (c) the combined effects of agricultural emissions and LULCC.**

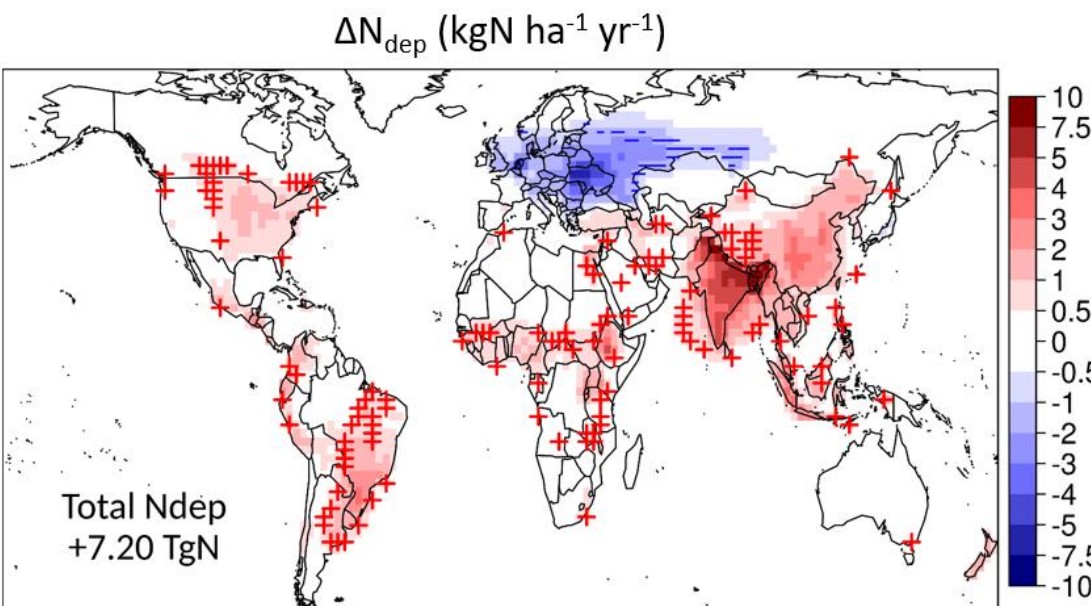

**Figure 7. Changes in total nitrogen deposition ($\Delta N_{dep}$) due to changes (1992-2014) in agricultural emissions and land cover. Red plus signs (+) mark new gridcells where total nitrogen deposition exceeds 5 kg N ha-1 yr-1, while blue minus signs (-) denote the gridcells where total nitrogen deposition decreases to below 5 kgN ha$^{-1}$ yr$^{-1}$.**