# Peer review of "Examining the competing effects of contemporary land management vs. land cover changes on global air quality"

_Atmospheric Chemistry and Physics, 2021_

## Author Comment (AC1)

We thank the reviewer for their comments related to the preprint discussion. If we understand correctly, we expect the reviewer may have additional comments for us to address in a full revision at a later stage.

We provide our responses to the quick preprint comments here, and we look forward to any additional comments/suggestions the reviewer may have for a full revision (at which stage we will of course provide a line-by-line response to all the comments in addition to these below).

*1) According to Figure 6, there is ozone change over the Atlantic ocean. Why is that?*

We thank the reviewer for this interesting question. We have checked the boundary layer ozone budget diagnostic over the region, and found that transport processes are largely responsible for the ozone change over the southern Atlantic Ocean. The ozone change in that region is the strongest during MAM (shown in Figure R1 below) and the weakest during JJA.

We note that similar transport may contribute to very small $O_3$ changes in other parts of the world, but our choice of colorbar in the manuscript purposefully de-emphasizes these exceedingly small effects (< 0.25 ppb) since they would be highly uncertain.

We will include this explanation in our revised manuscript during the full review stage (assuming we understand correctly that the reviewer may have additional comments that we will need to address in a full manuscript revision).

[Figure]

**Figure R1: Changes in MAM mean surface ozone (ppb) over southern Atlantic Ocean and the surrounding regions.**

*2) According to Table 2 and Figure 1. The changes in coverage is not unitless it's in percentage? The figure doesn't seem to be consistent to Table 2.*

We thank the reviewer for pointing out this potential source of confusion.

In Figure 1 our goal is to demonstrate and discuss the spatial pattern of land use and land cover changes. We choose to present the changes in coverage of each individual land type relative to

the area *per individual grid cell* (and as such these fractions are unitless). To clarify, these changes will be identical to percentages within that grid box: For example, a +0.1 "ΔNeedleleaf Forest" in Figure 1 indicates that needleleaf forests now occupy 10% more of the total grid cell area relative to the baseline.

For Table 2, we are summarizing land use and land cover changes in terms of the *global* total area covered by each land type. The "percentage changes" in column 4 are relative to the global total area covered by each individual land types at 1992.

We realize that we have used "coverage" to refer both the "fraction of area covered by a land type" (caption of Fig. 1) and "total area covered by a land type" (headings of Table 2), and appreciate the reviewer bringing up this source of confusion.

In our revised manuscript, we will change the caption of Figure 1 from ("…characterized by the changes in **coverages** (unitless)…", line 779) to "…characterized by the changes in grid box **fractional coverages** (unitless)…" for clarification in the revised manuscript. We welcome additional suggestions from the reviewer to further clarify this confusion.

We look forward to any additional comments from the reviewer for a full revision, and will of course provide a line-by-line response to all of their concerns with a fully revised manuscript that includes addressing these quick comments brought up in preprint discussion phase.

Thank you very much for your time,

Anthony Wong and Jeffrey Geddes

---

## Author Comment (AC2)

We thank the reviewer for their comments related to our manuscript. We provide a full response with changes to our manuscript below:

**Comments**

*1. According to Figure 6, there is ozone change over the Atlantic ocean. Why is that?*

Response: We thank the reviewer for this interesting question. We have checked the boundary layer ozone budget diagnostic over the region, and found that transport processes are largely responsible for the ozone change over the southern Atlantic Ocean. The ozone change in that region is the strongest during MAM (shown in Figure R1 below) and the weakest during JJA.

We note that similar transport may contribute to very small $O_3$ changes in other parts of the world, but our choice of colorbar in the manuscript purposefully de-emphasizes these exceedingly small effects ($< 0.25$ ppb) since they would be very uncertain.

[Figure]

**Figure R1: Changes in MAM mean surface ozone (ppb) over southern Atlantic Ocean and the surrounding regions.**

In response to the reviewer's comment, we have made the following changes in our manuscript:

**L 326: …(Fig. S6). Small surface $O_3$ changes, mainly due to transport, are also simulated over the Atlantic Ocean.**

2. *According to Table 2 and Figure 1. The changes in coverage is not unitless it's in percentage? The figure doesn't seem to be consistent to Table 2.*

Response: We thank the reviewer for pointing out this potential source of confusion. In Figure 1 our goal is to demonstrate and discuss the spatial pattern of land use and land cover changes. We choose to present the changes in coverage of each individual land type relative to the area per individual grid cell (and as such these fractions are unitless). To clarify, these changes will be identical to percentages within that grid box: For example, a +0.1 "ΔNeedleleaf Forest" in Figure 1 indicates that needleleaf forests now occupy 10% more of the total grid cell area relative to the baseline.

For Table 2, we are summarizing land use and land cover changes in terms of the global total area covered by each land type. The "percentage changes" in column 4 are relative to the global total area covered by each individual land types at 1992.

We realize that we have used "coverage" to refer both the "fraction of area covered by a land type" (caption of Fig. 1) and "total area covered by a land type" (headings of Table 2), and appreciate the reviewer bringing up this source of confusion.

In response to the reviewer's comment, we have made the following changes in our manuscript:

**L 779: … characterized by the changes in fractional coverage within a grid box (unitless)…**

---

## Author Comment (AC3)

*Wong and Geddes present work comparing the relative influence of land use/land cover change (LULCC) and agricultural reactive nitrogen emissions on air quality over modern timescales. They carry out this work using the GEOS-Chem chemical transport model along with a variety of updated emissions inventories and satellite products. In the end, the find that both effects can be important for regional air quality and trends in both LULLC and reactive nitrogen emissions should be considered when assessing multi-decadal trends in air quality. The manuscript is generally well structured and describes a thorough investigation. I can likely recommend this paper for publication after the following minor points are addressed.*

**Major Comments**

1. *The large number of specific geographical regions referenced in the manuscript substantially reduces the readability of the work. For example, how do the changes over Myanmar track with the changes across Southeast Asia (Germany and Benelux, Southern Russia and Kazakhstan, Southern Amazonia and Paraguay, etc.)? I suggest the authors standardize locations more consistently if possible.*

   Response: We thank the reviewer for making this suggestion to improve readability. When we describe the geographic patterns of LULCC, there are a lot of detailed sub-regional patterns that we thought might warrant certain specific geographic designation. We then introduced standardized definitions of regions in Table S2 for describing the area- and population-weighted average changes in $O_3$ and $PM_{2.5}$. We understand that in many occasions this choice could have reduced the readability of the manuscript where it is unnecessary to include specific geographic descriptions. In our revised manuscript, we increase the use of the standardized definitions of regions in Table S2, and include more specific regional descriptions only where necessary. Furthermore, we remove the use of unnecessary regional abbreviations (e.g. "SEA", "Cam", "WAf") and replace them with the full names of each region, which we feel also improves the readability of the manuscript.

   In response to this reviewer's comment, we have made the following changes to the manuscript:

   **L 162: …providing basis for our subsequent analyses. We also provide definitions for geographical regions, which largely follow Integrated modelling of global environmental change (IMAGE) 2.4 classifications, in Table S2.**

   **L 168 – 174: …cover types. Expansion of agricultural land at the expense of broadleaf forest coverage is most notable in  South America and Southeast Asia, which is well-documented…The expansion of agricultural land over this time period is also observed in central Asia,  Australia,  southern China and  Africa…built-up areas is observed in northern China and  Europe, consistent with the findings of Potapov et al. (2015) and Lai et al. (2016).**

**L 176 – 181:** Figure 2 shows the global changes in 3-year (2012-2014 minus 1991-1993) annual mean LAI calculated from the GLASS LAI data set. Over  southern China and South America,  the area with regionally consistent deforestation experience general increase in LAI, while the opposite effect is observed in Sahel and Former Soviet Union  In Europe, LAI  increases in most  parts despite a fairly consistent retraction of agricultural land is observed over the whole Europe…

**L 190 – 194:** …local reductions in isoprene emissions are observed in parts of South America …We note that the decrease of isoprene emission simulated in Southeast Asia does not agree with the result from Silva et al. (2016)…

**L 202 – 204:** …The magnitude of changes in soil NO emission induced by LULCC is comparable to that in agricultural NO emissions inventory (see below) over certain regions (e.g. South America, Australia,  Africa). Relatively large increases in soil NO is simulated over western Africa …

**L 207 – 212:** Slight increases of $v_d$ are observed in China, India, Southeast US,  central America,  South America, Europe and southern Africa. In Southeast Asia $v_d$ decreases concurrently with deforestation and reduction in LAI…Likewise, despite deforestation observed further south  these losses are offset by strong increases in LAI so that $v_d$ increases by up to 0.1 cm s$^{-1}$.

**L 221 – 224:** The increases in agricultural emissions are most substantial over  South Asia, followed by China, parts of  Middle East, Southeast Asia and  South America, and to a less degree in  central America, North America,  and  Sahel. The sharpest decline of agricultural emissions is observed in  Europe  and Former Soviet Union, followed by milder declines  over Japan and Korea...

**L 244 – 248:** Over parts of South America and Southeast Asia  where isoprene emissions drop significantly due to deforestation… Indeed, over  India and  China, where our model suggests high levels of SNA aerosol precursors…

**L 255 – 257:** … PM$_{2.5}$ concentrations are also observed in the Middle East, North America, central America and South America.

**L 258 – 264:** The largest decreases (up to 2.1 μg m$^{-3}$) in annual mean PM$_{2.5}$ due to changes in agricultural emissions are simulated in central and eastern Europe and Former Soviet Union.

 Despite comparable reductions in agricultural $NH_3$ emissions, decreases in $PM_{2.5}$ over **western Europe**  are smaller…agricultural emission changes simulated over western Europe are weaker than over **central and eastern Europe and Former Soviet Union.**

L 268 – 269: …due to agricultural emissions changes over  China and  India…

L 273: …We find that surface $PM_{2.5}$ over US, Europe and Former Soviet Union …

L 283 – 284: …observed over less populated areas.

L 298 – 290: … Regionally, the largest impact of land change ($\Delta PM_{2.5, \text{LULCC+agr\_emis}}$) on population-weighted annual mean surface $PM_{2.5}$ is simulated over **central and eastern Europe (CEU, -1.01 μg m$^{-3}$), Former Soviet Union (FSU, -1.00 μg m$^{-3}$), South Asia (SAs, +1.71 μg m$^{-3}$) and China (+1.45 μg m$^{-3}$)** …

L 292 - 298: The only exception to this occurs over North America  where anthropogenic NOx and $SO_2$ emissions have declines, but agricultural emissions have increased. This suggests that the increase in agricultural emissions over **North America**  has partially…on the order of 5% to 12% of changes due to direct anthropogenic emissions (e.g. in  **central and eastern Europe and western Europe** ). Notably, over **Former Soviet Union** , the Middle East  and central America …

L 310 – 311: Particularly, over the regions experiencing rapid change in land use intensity (e.g. **Former Soviet Union** ) or slow change in anthropogenic emissions (e.g. **central America, the Middle East** )…

L 317: …Over **parts of North America and central America** , the increase in dry deposition velocity…

L 320: …modelled surface ozone increases by up to 1.2 ppbv **further south** …

L 323 – 324: …However, in other parts of  Africa, up to 0.6 ppbv of surface ozone...

L 329 – 333: An exception to this is observed in the large increase in agricultural NOx emissions  which reduce surface $O_3$ by up to 0.6 ppbv over

NOx-saturated  India and  China, but increase surface $O_3$ in NOx-limited **parts of Southeast Asia**  by similar magnitude. Slight increases in surface $O_3$ level due to increased agricultural NOx emissions are also simulated over **parts of**  **eastern** Africa and **South America** . Whether the effect of agricultural emissions strengthens (e.g.  China and Sahel) or offset**s** (e.g. over southern Brazil and  India) **the effect of LULCC** is largely region-dependent.

L 348 – 353: Over Eastern Africa , Western Africa  and Southern Africa , area-averaged  $\Delta O_{3,\ LULCC+agr\_emis}$ generally has similar magnitudes to population-weighted  $\Delta O_{3,\ LULCC+agr\_emis}$. In other regions, the differences between area and population-weighted $\Delta O_{3,\ LULCC+agr\_emis}$ are more substantial. The largest discrepancies between area and population-weighted $\Delta O_{3,\ LULCC+agr\_emis}$ is found over China, where increases in surface $O_3$ are predicted over less populated western China, while reductions in surface $O_3$ are simulated over more densely-populated eastern China. In South America …

L 356 – 360:  **Over China, western Africa, eastern Africa, southern Africa, Former Soviet Union and the Middle East,** the magnitudes of population-weighted $\Delta O_{3,\ LULCC+agr\_emis}$ are more than 20% of that of $\Delta O_{3,\ anth}$, implying that contemporary land system changes could be a regionally important component in contemporary trends of surface $O_3$. The effects of agricultural emission changes and LULCC can either noticeably enhance (e.g. over **the Middle East** , **Japan and Korea** , China) or offset (e.g. over **South Asia** ) each other because…

L 380 – 388: The increase is mostly simulated over the Americas, Africa, **the Middle East**  and China, which is partially offset the large decrease over **Former Soviet Union** . Meanwhile, despite agricultural changes that lead to notable $\Delta N_{dep}$, over most of Europe,  South Asia and Southeast Asia, nitrogen input from other sources are large enough that this signal alone does not lead substantial changes in $N_{dep}$ exceedances of 5 kgN ha$^{-1}$ yr$^{-1}$. However, over **parts of North America, South America, Africa and China,**  agricultural changes are simulated to increase $N_{dep}$ from below to above 5 kgN ha$^{-1}$ yr$^{-1}$. This implies these natural ecosystems  of these areas are at risk of nitrogen exceedances due to agricultural changes. In contrast, the substantial reduction of $N_{dep}$ in **parts of Former Soviet Union**  may have significantly reduce the risk of nitrogen exceedance of natural ecosystem from agricultural sources.

L 400: …LULCC over  India and  China…

L 406 – 411: …Noticeable changes (> 1 µg m$^{-3}$) population-weighted $\Delta PM_{2.5,\ LULCC+agr\_emis}$ are simulated over China (+1.45 µg m$^{-3}$),  **South Asia** (+1.71 µg m$^{-3}$),  **central and eastern Europe** (-1.00 µg m$^{-3}$) and  **Former Soviet Union** (-

1.01 μg m$^{-3}$), indicating the potential impact of land change on long-term public health through modulating PM$_{2.5}$ level at regional scale. Our results suggest that contemporary (1996-2014) changes contribute to changes in PM$_{2.5}$ at regional and global scales that range from on the order of 5 to 10% of changes in PM$_{2.5}$ resulting from direct anthropogenic emissions over the same time period, and up to ~25% or more in Former Soviet Union, the Middle East and central America  specifically.

L 417 – 421: The increase in agricultural emissions reduces O$_3$ over NOx-saturated parts of China and South Asia  by… enhancements of dry deposition reduce O$_3$ over parts of China, North America and South America  by up to 1.2 ppbv. Overall, the largest population-weighted ΔO$_{3, LULCC+agr\_emis}$ is simulated over western Africa  (+0.42 ppbv) and eastern Africa  (+0.47 ppbv)…

2. *Following the text on lines 152-162 and the supplement, the GEOS-Chem model appears to have reasonably large issues in the simulation of SNA. Annual mean biases of 30-50% are not necessarily consistent with the model capturing "the present-day annual means of surface SNA" as stated on Line 161. The authors should describe how these biases influence the interpretation of the results in this work (e.g., are biases in annual magnitudes sufficiently unimportant for the simulation of changes in SNA?).*

Response: We thank the review for raising the important issue of model performance. From Fig. S1 we can see that GEOS-Chem reasonably captures the global geographic distributions of individual SNA species, but with biases in the absolute magnitudes.

As discussed in line 272 – 276, we find that the sensitivity of SNA to agricultural emissions does vary with anthropogenic emissions, and, by extension, to background SNA concentration. Therefore, against observations in US, China, and EANET-covered regions, we recognize that the model may have underestimated the sensitivity of SNA to agricultural emissions in our scenario. The effect of biases against the Canadian network data are harder to interpret, but the changes in PM$_{2.5}$ over Canada are generally small. To address the reviewer's concern, we have rewritten line 155 – 159 of our main text as follows:

L 155 – 159: … the same time period. ~~In general, the model captures the regional annual means of individual SNA species reasonably (Fig. S1 and Table S1), especially over US and Europe, where the bias is within ±30%. The model underestimates all SNA species over China in a relatively uniform fashion (36 – 55%). Over the region covered by Acid Deposition Monitoring Network in East Asia (EANET) (Japan, Korea and southeast Asia) the model underestimates the negatively charge ions (36% for sulphate and 16% for nitrate) while overestimating ammonium by 14%.~~ In general, the model captures the spatial distributions of individual SNA species reasonably well (Fig. S1). The model is able to capture

**regional annual means of individual SNA species (Table S1) over Europe. Over the US and China, where annual means of all SNA species are underestimated by 21 – 55%, and in regions covered by Acid Deposition Monitoring Network in East Asia (Japan, Korea and southeast Asia) where $SO_4^{2-}$ is underestimated by 36%, we expect the model mayunderestimate the sensitivity of SNA concentration to $NH_3$ emission perturbations. This may imply that results from our study should be interpreted as conservative. Figure S2…**

3. *Despite the nonlinearity in the response of atmospheric composition to changes in surface fluxes, the changes in ozone and PM due the combined effects of agricultural emissions and LULCC (Tables 3 and 4) are nearly linear with respect to the individual process changes. Do the authors have any hypotheses as to why this might be?*

   Response: We thank the reviewer for noticing this fundamental question. We had hypothesized that the changes in BVOC emissions due to LULCC, and $NH_3$ emissions due to agriculture, might be strong enough to change the chemical regime of $O_3$ and SNA production. Therefore, we would not have ruled out the possibility of non-linear interactions between these two factors, which may indeed be present over different time scales. Yet as the reviewer notes, the effects of agricultural emissions and LULCC are approximately linearly additive over our period of interest. We expect this is mainly because of two reasons: (1) the changes in surface fluxes on this time scale are not large enough to change the $O_3$ and SNA chemical production regimes; and (2) LULCC mainly impacts $O_3$ precursors, while agricultural emissions mainly impact SNA precursors, which are in many cases spatially segregated. The interaction between these two factors in space turns out to be relatively small on the timescales investigated.

   In response to the reviewer's observation, we make note of this result in our revised manuscript:

   **L 442: …despite the relative small signal that we obtain here.**

   **We find the effects of agricultural emissions and LULCC to be largely linearly additive over contemporary timescales, which may be attributable to two factors: 1) LULCC mainly impacts $O_3$ precursors while agricultural emissions mainly impact SNA precursors, and these are often spatially segregated; 2) LULCC and agriculture-related changes in surface fluxes of $O_3$ and SNA precursors are not large enough to change their respective chemical production regime. At longer timescale when land change signals are stronger, the effects of LULCC and agricultural emissions may be non-linear.**

**Minor Comments**

1. *L23-25: This statement is sufficiently qualified to be nearly meaningless and could be much stronger. Your work does more than demonstrate possible impacts which imply potential importance!*

Response: We thank the reviewer for recognizing the importance of our results. We make the following change:

**L 23 – 25: Our results demonstrate the  impacts of contemporary LULCC and agricultural $N_r$ emission changes on $PM_{2.5}$ and $O_3$ air quality,  and the importance of land system changes on air quality over multi-decadal timescales.**

2. *L143-145: Are there other LULCC impacts on meteorology which the authors think might important that aren't addressed through changing the roughness length?*

Response: We thank the reviewer for this theoretically important question, which can be clarified in two different directions. In terms of surface exchange schemes for chemical transport models, changes in canopy heights, and therefore displacement height ($h$), might also have an impact on aerodynamic resistance ($R_a$) in addition to roughness length ($z_0$) (the latter of which is considered in our study). However, the former effects from canopy height are not considered by default in GEOS-Chem, and therefore neither in our modelling study. We hypothesize the effect of $h$ will be small, by considering how $R_a$ is typically calculated in land surface exchange scheme:

$$R_a = \frac{1}{\kappa u_*} (\ln\left(\frac{z-d}{z_0}\right) - \Psi\left(\frac{z-d}{L}\right) + \Psi(\frac{z_0}{L}))$$

Since the middle of the first vertical grid of GEOS-Chem ($z$) is around 60 – 70 meters (http://wiki.seas.harvard.edu/geos-chem/index.php/GEOS-Chem_vertical_grids), which is significantly larger than $d$ such that $z - d \approx z$, the changes in $d$ are not nearly as important as the changes in $z_0$. We include this argument in our supplemental material

Another important dimension is how LULCC directly impact mesoscale and large-scale meteorology through changing sensible heat and latent heat fluxes, and its subsequent influence on transport and chemistry. Similarly, agricultural emissions may also perturb meteorology mainly through aerosol and cloud radiative effects. While these effects can be important (e.g. Wang et al., 2020), assessing their importance requires comprehensive climate and Earth system modelling, was not in the scope of our manuscript but deserves to be mentioned.

To respond to the reviewer's comment, we have made the following these changes in our manuscript:

**L 143-145: …The dominant surface type can be readily mapped to the 11 deposition surface type in the Wesely dry deposition model. We adopt the approach of Geddes et al. (2016) to replace roughness length ($z_0$) from assimilated meteorology with that prescribed for each deposition surface type. We ignore changes in in displacement height as they are expected to be much less important than the changes in $z_0$ (Text S1). To derive…**

**L 454-455: …computed LAI changes and trends, and these have been shown to be important to changes in simulated $O_3$ in this study and elsewhere (Wong et al. 2019). Finally, the meteorological feedbacks (e.g. changes in sensible heat, latent heat, air temperature, boundary layer height) and the subsequent effects on atmospheric chemistry and transport from LULCC and agricultural emissions are not considered in our study, which could potentially be important** (e.g. Wang et al., 2020)**.**

**Text S1. Considering how aerodynamic resistance ($R_a$) is typically calculated in land surface exchange scheme:**

$$R_a = \frac{1}{\kappa u_*}(\ln\left(\frac{z-d}{z_0}\right) - \Psi\left(\frac{z-d}{L}\right) + \Psi(\frac{z_0}{L}))$$

**where κ is von Kármán constant, $u_*$ is friction velocity (m s[-1]), $L$ is Obukhov Length ($L$) and $d$ is displacement height (m). Since the middle of the first vertical grid of GEOS-Chem ($z$) is around 60 – 70 meters (http://wiki.seas.harvard.edu/geos-chem/index.php/GEOS-Chem_vertical_grids), which is significantly larger than $d$ such that $z - d \approx z$ under most conditions, the changes in $d$ sare expected to be less important than the changes in $z_0$.**

3. *L183-186: What is the potential size and influence of this effect on the results in this work?*

   Response: We thank the reviewer for the important question. Accurately assessing this questions requires comprehensive comparisons between LAI retrieval from static land cover map versus that from dynamic land cover map. A quantitative study may be outside of the scope of our study, but the question deserves to be addressed in our manuscript. Based on the work of Fang et al. (2013), we provide the following argument for the potential size and influence of such effect, to address the reviewer's question:

   *L183-186:* **… land use change. We note that since the relationship between satellite-derived surface reflectance and retrieved LAI depends on land cover, the use of static land cover map in long-term LAI retrievals (Claverie et al., 2016; Xiao et al., 2016; Zhu et al., 2013) may not fully capture the effect of LULCC on LAI (Fang et al., 2013). In particular,** Fang et al. (2013) **show that LAI could be substantially overestimated when grasses and crops are misclassified as forest. We may therefore overestimate dry deposition velocity over regions with significant deforestation. Such impact on biogenic emissions is secondary as biogenic emissions are typically much more sensitive to land cover type than LAI** (e.g. Guenther et al., 2012)**.**

4. *L195-196: This seems like a bigger issue than just in Southeast Asia as it relates to oil palm plantations. Presumably everywhere that relatively large land cover changes occur that are not represented in the datasets used here will be missed.*

Response: We thank the reviewer for the important question. Southeast Asia is a special case as palm plantations are very widespread in that region, and they have much higher isoprene emission than other evergreen broadleaf trees, creating a systematic bias that contradicts with the common understanding of how tropical deforestation reduces isoprene emissions. This phenomenon is relatively well-documented (Silva et al., 2016; Stavrakou et al., 2014).

It is true that intra-PFT variabilities of BVOC emissions factors can be large, although the use of PFT-based emission factor in regional and global modelling is generally justifiable (Guenther et al. (2012). We suppose this could may be of importance in other examples of LULCC that we may not be aware of, so to address the reviewer's comment, we have added this further caveat in our text:

**L 455: …Though the use of PFT-based emission factors in regional and global modelling is generally justifiable** (Guenther et al., 2012)**, we cannot rule out the possibility of intra-PFT variabilities of BVOC emission factors affecting the accuracies our results, which is exemplified by the inability of our model to capture the palm-driven isoprene emission increase over Southeast Asia** (Silva et al., 2016) **as discussed in section 3.**

*There are minor grammatical errors throughout the manuscript, related dominantly to article use and subject-verb agreement. Some of these are listed below:*

*L11 "cause reduction" to "cause a reduction", "level" to "levels"*

*L12 "level India, China and eastern US" to "levels in India, China and the eastern US"*

*L14 "Across" to "across"*

*L35 "...introduce an enormous amount"*

*L340 "likely"*

Response: We have made all revisions as suggested

---

## Referee Report (RR1)

Apologies to the late reply. These are the comments to the revised version
https://doi.org/10.5194/acp-2021-132-AC2

The scope of this manuscript is broad, as it is about the relationship between land use change and air pollutants emission, it is relevant in a global scale. The research question is important for interdisciplinary interest and has real world implication. The research has a high potential for societal and policy impact.
Wong and Geddes use a factorial experiments approach to investigate the relative impact of Agricultural emissions, LULCC and Anthropogenic emissions. Wong and Geddes found that agricultural emissions have strong effects on PM2.5, while LULCC has strong effects on $O_3$.

Thanks a lot for consolidating the comments into a revised manuscript. It is much better now especially after addressing the comments from Reviewer 2.

Here are some specific comments for your consideration.

1. I feel that the title of the paper does not sound right. Perhaps it could be "Study on the competing effects …" Instead of starting with "On" .
2. There are many complex sentences that I think it is too long. Try split those long sentences into several short one, it would improve the readability.
3. The Introduction is a bit short; it would be helpful if you could include more detail about how agricultural emissions lead to the formation of secondary PM2.5. Under what conditions that would affect its formation. Same for $O_3$ as well.
4. Line 67: Any better word usage other than "contemporaneously"?  How about simultaneously?
5. Line 89: "an" instead of "a"
6. Line 93: Why use "fifth" here instead of "(5)", be consistent.
7. Line 145: "in"
8. Line 179: area with regionally consistent deforestation experience increase in LAI? I thought it would decrease LAI when turning forest into grass, please explain.
9. Line 183: LAI increases in northern China is not because of decrease of agricultural land but could be afforestation and turning desert to farmland? You can search for the keywords: Green wall of China. And this Nature paper
https://www.nature.com/articles/s41893-019-0220-7
10. Line 227: what causes the sharp decline of agricultural emissions in Europe? Reduce farming activity? Implementation of clean air policies?
11. Line 235: Could be explain by afforestation. See point 9.
12. Line 272: I still do not understand clearly how you calculate the population-weighted average. Do you mean per capita? Where do you get the population data from?
13. Line 281: "land change phenomena", do you mean land use change? Or the area change as what Table 3 suggests? Please use a word that would not confuse the readers.
14. Conclusions: It is good that you mentioned about the limitation in the study. You could also add several sentences to discuss about the implication on policies. What policies could help reduce or mitigate the impact of LULCC, agricultural and anthropogenic emissions? How do you prioritise it? It will increase the impact of your paper to policymakers.

15. The caption of the figure should allow readers to understand the figure without looking at the main text, self-explanatory. If possible, write the complete form of acronyms that are not used frequently in the main text, for example CEDS in Figure 4. If space allows, I will write "Leaf Area Index" instead of LAI in Figure 2. Same for Figure 5b and 6b, you could write "Agricultural Emissions effect" instead.

---

## Editor Decision (ED1)

**Review of paper acp-2021-132: On the competing effects of contemporary land management vs. land cover changes on global air quality by Wong and Geddes**

Dear author, co-authors,

First of all, sorry for the delay in my decision on your manuscript having received already some of the last reviewers comments quite a long time ago. Anyhow, I didn't want to take a hasty decision and needed to the find the time to carefully check once more again the reviews and your response to the shared feedback. This was a good decision since in doing so, also reading over again the revised ms I came across a number of more minor issues, that should be anyhow resolved. But there are also still some more major issues that anyhow must resolved and some that you could potentially consider in a further revision. This mainly refers to some specific features of the representation of atmosphere-biosphere exchange in the GEOS-CHEM modelling system you applied for your analysis and which have not been raised by the reviewers.

Lines 39:40; remove the double point at begin of sentence there and also refer consistently to $NO_x$ (subscript x).

Line 53: "in land cover .". By putting the term classification you make it appear more that the classification of land cover is the main issue where it is simply "no changes in land cover"

Lines 70:74: I do really appreciate your point about the fact that we need to consistently consider the combined effect of both LULCC and changes in agricultural emissions also since here we might see some compensating effects. This was actually one of the main take-home messages also of the Ganzeveld et al. 2010 study in which we included in the most consistent manner these combined effects but then in a study on the anticipated future changes in LULCC and agricultural N-emissions (only NOx). I am very much aware that by sharing this comment at this stage that 1) I should have done this in an earlier stage and 2) that I am really in doubt making this comment since I don't want to leave any impression of "pushing" my own papers. But now having read again this particular statement making a strong point about this consistent representation of all involved processes, I bring it up since I also see that this is actually a shortcoming of many other studies. This is also further stressed in the follow-up statements in lines 76-79.

Chapter 2: Methods; The description of all the steps to consider the dependence of dry deposition and emissions on LULCC and agriculture makes clear that you made a large effort to consistently consider the impact of this on these two processes. However, it triggers the question to what extent your results might then be missing one specific aspect of atmosphere-biosphere exchange that might be quite important for the overall/compensating effects; canopy interactions; e.g., how much of the emitted NOx and NH3 is really escaping the vegetation canopy (especially relevant for large LAI's) and how a decoupled treatment of soil-canopy N-emission and deposition would further effect your results. The first feature, also referred to as the canopy reduction factor is considered in the Hudman et al. soil NOx emission inventory but how did you handle this for NH3? And how for the fertilizer-application driven NOx emissions in your approach (reading that those were removed from the Hudman inventory to avoid double counting). In addition, there would be some other aspects of atmosphere-biosphere exchange of relevance for your study, and that should be included in the discussion: how does the deposition representation in GEOS-CHEM consider the dependence on stomatal exchange and soil water status (an important feature of LULCC). You refer in the discussions shortly to the fact that e.g., the

coupling with latent heat exchange and boundary layer dynamics has been ignored. I am very much aware that some of these features (and uncertainties) in LULCC and agricultural management are likely much more important but not having considered these additional dependencies of the system in a consistent manner is important to indicate already at an earlier point in your ms. In addition, there are other aspects of (N) atmosphere-biosphere exchange that have not been mentioned at all and might be quite relevant, existence of NOx and NH3 compensation points.

Properly discussing these potentially important features is required also reading lines 211-213: "represent the change in soil emission driven purely by LAI and land cover changes"

The same holds for the statement in line 223-224: "Significant changes in the vd of O3 due to LAI also imply that vd of other relevant trace gases (e.g. NO2, SO2)"; how is the deposition of $NO_2$ being treated in GEOS-CHEM, e.g., does it consider a significant N-compensation point for ecosystems prone to high N loading?

Line 305: "have declined"

Lines 323-325: "the impacts of agricultural emission changes on O3 ("ΔO3, agr_emis") is the difference in  $O_3$ predicted by Simulation 3 and Simulation 2; and the impacts of these combined ("ΔO3, LULCC+agr_emis") is the difference in  $O_3$ predicted Simulation 3 and Simulation 1"

Lines 333-334: Here there is an apparent flaw: "In contrast, modelled surface ozone increases by up to 1.2 ppbv further south, where strong increases in LAI lead to largely increases vd"; $O_3$ increasing due to enhanced dry deposition? It should also read as "lead to large increases in $v_d$" and what is large? Give a percentage or the absolute numbers.

Line 336: "up to 0.6 ppbv of surface ozone increases are simulated, mainly because of the relatively large increase in soil NO emission". This is an example that triggers the question what happened to the effective emissions into the atmosphere; is it indeed purely the changes in the soil NO emissions (due to temperature or moisture effects, or management) and how much an effect is there by changes in the canopy reduction factor due to changes in LAI?

Line 341: "NOx -> $NO_x$" and check this for consistency.

Line 367: "but  there the positive and negative largely offset each other"

Line 395: "which partially offsets"

Line 397: "does not lead to substantial changes"

Line 423: "Our results suggest that contemporary (1996-2014) changes in LULCC and agricultural emissions contribute to changes.

Lines 428-230; these conclusions are consistent with the findings by Ganzeveld et al. on the small impact of future LULCC and agricultural emissions changes on ozone also due to a number of compensating effects. I think it would be very useful to stress that your findings on contemporary versus future changes in LULCC and agricultural emissions in different modelling systems/approaches come up with such a consistent finding.

---

## Author Response (AR2)

*Apologies to the late reply. These are the comments to the revised version*
[https://doi.org/10.5194/acp-2021-132-AC2](https://doi.org/10.5194/acp-2021-132-AC2)
*The scope of this manuscript is broad, as it is about the relationship between land use change and air pollutants emission, it is relevant in a global scale. The research question is important for interdisciplinary interest and has real world implication. The research has a high potential for societal and policy impact.*
*Wong and Geddes use a factorial experiments approach to investigate the relative impact of Agricultural emissions, LULCC and Anthropogenic emissions. Wong and Geddes found that agricultural emissions have strong effects on PM2.5, while LULCC has strong effects on O3.*
*Thanks a lot for consolidating the comments into a revised manuscript. It is much better now especially after addressing the comments from Reviewer 2.*
*Here are some specific comments for your consideration.*

**Comments**

1. *I feel that the title of the paper does not sound right. Perhaps it could be "Study on the competing effects …" Instead of starting with "On" .*

    Response: We thank the review for this suggestion. We change our title to:

    **Examining the competing effects…**

2. *There are many complex sentences that I think it is too long. Try split those long sentences into several short one, it would improve the readability.*

    Response: We thank the review for this suggestion. In response to this reviewer's comment, we have made the following changes to the manuscript:

    **L 76 – 77:**  Consistent long-term  records of land cover derived from satellite remote sensing observations and global anthropogenic emission inventories have become readily available. This opens an opportunity…
    **L 173 – 174:** …have increased mainly at the expense of forest coverage.  This is consistent with a global trend in deforestation over this period…
    **L 203 – 204:** … The largest local reductions in isoprene emissions (up to 30%) are observed in parts of South America, where deforestation from highly isoprene-emitting broadleaf forests is most strongly observed.
    **L 406 – 410:** …We model the effects of contemporary LULCC and agricultural emission changes, individually then in combination, on surface $O_3$ and $PM_{2.5}$ using the GEOS-Chem CTM. With a uniquely consistent framework,  we are able to integrate direct information from global emission inventories (CEDS) with updated land surface remote sensing products (ESA CCI land cover and GLASS LAI).  This allows us to avoid invoking extra

**assumptions on land management practices (e.g. constant Nr input, emissions or emission factors over time) and biophysical properties of PFTs (e.g. constant PFT-specific LAI over time).**

3. *The Introduction is a bit short; it would be helpful if you could include more detail about how agricultural emissions lead to the formation of secondary PM2.5. Under what conditions that would affect its formation. Same for O3 as well.*

Response: We thank the review for this suggestion. We agree that this will increase the readability of our manuscript. In response to this reviewer's comment, we have made the following changes to the manuscript:

**L 36 – 42: The reactive nitrogen oxides emitted from soil, NOx ($\equiv$ NO + NO$_2$), is a key component of O$_3$ photochemistry enhance O$_3$ production when volatile organic compounds (VOCs) are relatively abundant (i.e. NOx-limited regimes), but suppresses O$_3$ production when the concentration of VOCs is relatively low (i.e. VOC-limited regimes) (Sillman et al., 1990). Reactive nitrogen also contributes to aerosol formation. Ammonia (NH$_3$) can combine with the nitrate and sulphate ions to form secondary inorganic aerosol, while the emissions of NOx can oxidize further and contribute to particulate nitrate formation (Ansari and Pandis, 1998). Indeed, agricultural emissions are the dominant global anthropogenic source of ammonia (NH$_3$) (Hoesly et al., 2018)…**

4. *L 67: Any better word usage other than "contemporaneously"? How about simultaneously?*
5. *L 89: "an" instead of "a"*
6. *L 93: Why use "fifth" here instead of "(5)", be consistent.*
7. *L 145: "in"*

Response: We have made all revisions above as suggested.

8. *L 179: area with regionally consistent deforestation experience increase in LAI? I thought it would decrease LAI when turning forest into grass, please explain.*

   *L 183: LAI increases in northern China is not because of decrease of agricultural land but could be afforestation and turning desert to farmland? You can search for the keywords: Green wall of China. And this Nature paper https://www.nature.com/articles/s41893-019-0220-7*

   *L 235: Could be explain by afforestation. See point 9.*

Response: We thank the reviewer for the three related comments above. We appreciate the suggestion of an excellent reference about LAI changes in general, particularly about India and China. After careful studying, we find the paper suggesting in addition to reforestation, greening (increase of LAI) within cropland and forests also contributes to overall greening.

In response to this reviewer's comments, we have made the following changes to the manuscript:

**L 188 – 192: …****For example, the general increase of LAI in China is not only driven by changes in biome types, but also the greening within cropland (mainly attributable to agricultural intensification) and forests (mainly attributable to ambitious tree planting programmes) (Chen et al., 2019). Similarly, some deforested land in South America might have been cultivated intensively, resulting in an increase rather decrease in LAI.** **We** **also** **note that since the…**

**L 245 – 246: …over this same period.** **Such agricultural intensification in turn contribute significantly to the positive LAI trend over the above regions (Chen et al., 2019).** **Similarly, agricultural emissions…**

9. *L 227: what causes the sharp decline of agricultural emissions in Europe? Reduce farming activity? Implementation of clean air policies?*

   Response: We thank the reviewer for their questions. We agree that the decline of NH$_3$ emissions in Europe is indeed very significant, meanwhile well-documented and analyzed, which warrant deeper discussion.

   In response to this reviewer's comment, we have made the following changes to the manuscript:

   **L 235 – 237:** **The particularly sharp decline of agricultural emissions in Europe is mainly attributable to the implementation of emission control protocols (National Emissions Ceilings (NEC) and Integrated Pollution Prevention and Control (IPPC) directives) within the European Union (Skjøth and Hertel, 2013).** **According to the CEDS inventory…**

10. *L 272: I still do not understand clearly how you calculate the population-weighted average. Do you mean per capita? Where do you get the population data from?*

    Response: We thank the reviewer for pointing out the ambiguity. In response to this reviewer's comment, we have added description of our method of calculating population-weighted average in supplemental material:

    **Text S2.** The population-weighted averaged changes surface O$_3$ (ppb) or PM$_{2.5}$ (µg m$^{-3}$) ($\Delta[X]_{\text{pop\_weighted, Y}}$) for region Y is calculated as follow:

    $$\Delta[X]_{\text{pop\_weighted,Y}} = \frac{\sum_i^{gridcells\ in\ Y} \Delta[X]_i Pop_i}{\sum_i^{gridcells\ in\ Y} Pop_i} (S2)$$

    where $\Delta[X]_i$ is changes in surface concentration of concerned chemical species, and *Pop$_i$* is the population count for individual gridcell *i*. The global gridded population is from the fourth version of The Gridded Population of the World (GPWv4) (CIESIN, 2018), and remapped to match the resolution of GEOS-Chem output.

We also make this change in our main text to reference the supplemental text:

**L 282: …so that the effects on population-weighted average (method described in Text S2)…**

11. *L 281: "land change phenomena", do you mean land use change? Or the area change as what Table 3 suggests? Please use a word that would not confuse the readers.*

Response: We thank the reviewer for pointing out the ambiguity. In response to this reviewer's comment, we have clarified our wording:

**L 291:  Table 3 summarizes the simulated effects of  LULCC and agricultural emission changes on PM$_{2.5}$…**

12. *Conclusions: It is good that you mentioned about the limitation in the study. You could also add several sentences to discuss about the implication on policies. What policies could help reduce or mitigate the impact of LULCC, agricultural and anthropogenic emissions? How do you prioritise it? It will increase the impact of your paper to policymakers.*

Response: We thank the reviewer for this constructive comment. We agree that some discussion about potential policy priorities to mitigate air pollution from land system is a valuable addition without sacrificing the scientific rigor of our manuscript.

In response to this reviewer's comment, we have made the following changes to the manuscript:

**L 488 – 496: Incentivizing these and other practices that improve agricultural nitrogen use efficiency (e.g. including livestock production with cropping, synchronizing nitrogen supply with crop demand) (e.g. Fageria and Baligar, 2005; Langholtz et al., 2021) can be one of the keys to mitigate the air quality impacts of reactive nitrogen input without compromising agricultural productivity (e.g. Guo et al., 2020). Furthermore, as increasing reactive nitrogen input and land use change are the two of the main strategies to meet the global demand for biomass-based products in the future (Foley et al., 2011), the distinct yet significant impacts of agricultural emissions and land use change on O3, PM2.5 and nitrogen deposition should be investigated as part of the overall environmental impacts of land system changes, especially when tradeoff between increasing land input and cropland expansion exists (e.g. Lotze-Campen et al., 2010; Mauser et al., 2015). This could benefit agricultural policy activities by appropriately considering all in the externalities and socioeconomic costs of different options and scenarios for agricultural expansion.**

13. *The caption of the figure should allow readers to understand the figure without looking at the main text, self-explanatory. If possible, write the complete form of acronyms that are not used frequently in the main text, for example CEDS in Figure 4. If space allows, I will write "Leaf Area Index" instead of LAI in Figure 2. Same for Figure 5b and 6b, you could write "Agricultural Emissions effect" instead.*

We thank the reviewer for this suggestion. In response to this reviewer's comment, we have made the following changes to our figure captions:

**L 824 – 825: Figure 2. Global Land Surface Satellite (GLASS) -derived changes in 3-year mean annual  leaf area index (LAI) (2012 to 2014 average minus 1991 to 1993 average).**

**L 830 – 831: Figure 4. Changes in agricultural $NH_3$ and $NO_x$ emissions (2014 – 1992) as implemented by the Community Emissions Data System (CEDS).**

**L 832 – 833: Figure 5. Simulated changes in annual mean surface $PM_{2.5}$ due to (a) LULCC, (b) agricultural emission ("Agr Emis") changes, and (c) the combined effects of agricultural emissions and LULCC.**

**L 834 – 835: Figure 6. Simulated changes in annual mean surface $O_3$ due to (a) LULCC, (b) agricultural emission ("Agr Emis") changes, and (c) the combined effects of agricultural emissions and LULCC.**

---

## Editor Decision (ED2)

**Review of paper acp-2021-132: On the competing effects of contemporary land management vs. land cover changes on global air quality by Wong and Geddes**

Dear author, co-authors,

I have checked your response to the comments I raised in my editors review and the revisions you included in the manuscript. It appears that overall you appreciated the feedback seeing the revisions. I now accept the paper for publication in ACP after you have handled some minor last issues I found in reading your response and the revision,

Line 140: "the fertilizer emissions  represent only.."

Lines 352:353; " In contrast, modelled surface ozone increases decreases by up to 1.2 ppbv further south, where strong increases in LAI lead to large increases in vd (up to 0.06 m s-1)"; An increase of 0.06 m s-1 is an increase of 6 cm s-1. That is indeed a large Vd given that the maximum for ozone is ~2 cm s-1.... If it is a typo and should be 0.06 cm s-1 than it is not a large change; a change of 0.6 cm s-1 would be indeed a large change. So what is it at the end?

Line 496: "Zhang et al., 2010) are" (space missing)

---

## Author Response (AR3)

Dear Handling Editor,

Many thanks for your careful review of our manuscript. We appreciate your time, and value your concerns. We believe we have addressed all your concerns in our revised manuscript. Below, we respond to each of your comments individually, followed by the appropriate changes to our manuscript.

We look forward to your timely response to this revision, and hope that you will agree our manuscript has improved in response to your input.

*First of all, sorry for the delay in my decision on your manuscript having received already some of the last reviewers' comments quite a long time ago. Anyhow, I didn't want to take a hasty decision and needed to the find the time to carefully check once more again the reviews and your response to the shared feedback. This was a good decision since in doing so, also reading over again the revised ms I came across a number of more minor issues, that should be anyhow resolved. But there are also still some more major issues that anyhow must resolved and some that you could potentially consider in a further revision. This mainly refers to some specific features of the representation of atmosphere-biosphere exchange in the GEOS-CHEM modelling system you applied for your analysis and which have not been raised by the reviewers.*

**Comments**

*Lines 39:40; remove the double point at begin of sentence there and also refer consistently to NOx (subscript x).*

*Line 53: "in land cover classification.". By putting the term classification you make it appear more that the classification of land cover is the main issue where it is simply "no changes in land cover"*

*Line 305: "have declined"*

*Lines 323-325: "the impacts of agricultural emission changes on $O_3$ ("$\Delta O_{3,\ agr\_emis}$") is the difference in $PM_{2.5}$ $O_3$ predicted by Simulation 3 and Simulation 2; and the impacts of these combined ("$\Delta O_{3,\ LULCC+agr\_emis}$") is the difference in $PM_{2.5}$ $O_3$ predicted Simulation 3 and Simulation 1"*

*Line 341: "NOx -> NOx" and check this for consistency.*

*Line 367: "but these there the positive and negative largely offset each other"*

*Line 395: "which partially offsets"*

*Line 397: "does not lead to substantial changes"*

*Line 423: "Our results suggest that contemporary (1996-2014) changes in LULCC and agricultural emissions contribute to changes.*

Response: We have made all revisions above as suggested.

*Lines 70:74: I do really appreciate your point about the fact that we need to consistently consider the combined effect of both LULCC and changes in agricultural emissions also since here we might see some compensating effects. This was actually one of the main take-home messages also of the Ganzeveld et al. 2010 study in which we included in the most consistent manner these combined effects but then in a study on the anticipated future changes in LULCC and agricultural N-emissions (only NOx). I am very much*

*aware that by sharing this comment at this stage that 1) I should have done this in an earlier stage and 2) that I am really in doubt making this comment since I don't want to leave any impression of "pushing" my own papers. But now having read again this particular statement making a strong point about this consistent representation of all involved processes, I bring it up since I also see that this is actually a shortcoming of many other studies. This is also further stressed in the follow-up statements in lines 76-79.*

Response: We agree with the editor that the conclusion of Ganzeveld et al. 2010 study is highly relevant. We have included a more detailed discussion in our manuscript:

**L 70 – 84: While changes in land cover and agricultural emissions actually occur simultaneously across the globe, they are rarely considered together in simulations of air quality from chemical transport models.** **The importance of studying these combined processes at the same time was highlighted by Ganzeveld et al. (2010) in their analysis of air quality impacts from future land use and land cover changes. In this study, for example, opposing effects on $O_3$ were simulated with decreases in tropical forest soil $NO_x$ emissions being compensated by increases in soil $NO_x$ emissions associated with agriculture. Still, this work did not explore the concomitant changes in ammonia emissions that would be expected with the changes in agricultural activity. It remains unclear** **to what extent LULCC… This opens an opportunity for a more holistic and observationally-constrained assessment of the impacts on global $O_3$ and PM air quality from contemporary changes in LULCC and agricultural emissions simultaneously,** **which has been advocated by Ganzeveld et al. (2010),** **and a comparison…**

*Chapter 2: Methods; The description of all the steps to consider the dependence of dry deposition and emissions on LULCC and agriculture makes clear that you made a large effort to consistently consider the impact of this on these two processes. However, it triggers the question to what extent your results might then be missing one specific aspect of atmosphere-biosphere exchange that might be quite important for the overall/compensating effects; canopy interactions; e.g., how much of the emitted NOx and NH3 is really escaping the vegetation canopy (especially relevant for large LAI's) and how a decoupled treatment of soil-canopy N-emission and deposition would further effect your results. The first feature, also referred to as the canopy reduction factor is considered in the Hudman et al. soil NOx emission inventory but how did you handle this for NH3? And how for the fertilizer-application driven NOx emissions in your approach (reading that those were removed from the Hudman inventory to avoid double counting). In addition, there would be some other aspects of atmosphere-biosphere exchange of relevance for your study, and that should be included in the discussion: how does the deposition representation in GEOS-CHEM consider the dependence on stomatal exchange and soil water status (an important feature of LULCC). You refer in the discussions shortly to the fact that e.g., the coupling with latent heat exchange and boundary layer dynamics has been ignored. I am very much aware that some of these features (and uncertainties) in LULCC and agricultural management are likely much more important but not having considered these additional dependencies of the system in a consistent manner is important to indicate already at an earlier point in your ms. In addition, there are other aspects of (N) atmosphere-biosphere exchange that have not been mentioned at all and might be quite relevant, existence of NOx and NH3 compensation points. Properly discussing these potentially important features is required also reading lines 211-213: "represent the change in soil emission driven purely by LAI and land cover changes" The same holds for the statement in line 223-224: "Significant changes in the vd of $O_3$ due to LAI also imply that $v_d$ of other relevant trace gases (e.g. NO2, SO2)"; how is the deposition of NO2 being treated in GEOS-CHEM, e.g., does it consider a significant N- compensation point for ecosystems prone to high N loading?*

Response: We thank the editor for highlighting the importance of considering other important processes in comprehensively simulating atmosphere-biosphere exchange.

The question of bidirectional exchange and compensation points is a very interesting one to consider, which we should have addressed more explicitly in our manuscript. We note that bidirectional exchange has in some cases been implicitly accounted for in the CEDS inventory for ammonia fertilizer emissions. For example, in CEDS, the agricultural ammonia emissions over the United States are scaled to the NEI emissions estimate, and would therefore reflect some of the assumptions included in the NEI emissions modeling for ammonia (which is based on an implementation of bidirectional exchange in the CMAQ model). However, we cannot necessarily comment with certainty on how this would be treated elsewhere, and this will introduce an element of uncertainty in our simulation. It is also the case that the current public release of GEOS-Chem otherwise does not have online bidirectional exchange parameterizations for any species.

In response to this comment from the editor, we discuss the importance of bidirectional exchange for $NH_3$, and evidence for atmospheric compensation points for $NO_2$, in our revised manuscript. We clarify that in some cases these effects may have been accounted for implicitly, but include the caveat that our work cannot account for all these effects everywhere, since it is not yet the state of the science for GEOS-Chem. We discuss how neglecting these effects might contribute to uncertainties in our simulation, by drawing on global studies that have implemented bidirectional exchange for ammonia, and elaborate on the importance of this in future model investigation.

Regarding canopy uptake, we clarify in our revised manuscript that indeed the biogenic soil NOx emissions calculated using the Hudman scheme includes an explicit online treatment of canopy uptake. The implicit accounting of canopy uptake in the agricultural emissions may again depend somewhat on whether CEDS has scaled the emissions to a particular regional inventory that accounts for this (as would be the case over the United States). For simplicity, therefore, we make the assumption that the CEDS inventory provides an estimate of "above canopy" emissions into the atmosphere. We further argue that these agricultural emissions from soil represent only a fraction of the total agricultural emissions considered here (e.g. in addition to manure- and waste-associated emissions), so that the uncertainty introduced by the canopy reduction factor is only applicable for a fraction (~35-50% depending on region) of the agricultural ammonia emissions, reducing somewhat the apparent importance of quantifying this. Likewise, the fertilizer soil NOx emissions represent a small overall fraction of the total global soil NOx emissions (which in our simulation otherwise include a canopy reduction estimate). In response to the editor's comment we clarify these points in our revised manuscript, and include a caveat that this deserves detailed attention in future implementations of globally consistent emissions inventory development.

Overall, we believe the editor has raised excellent points of concern which deserved to be addressed in our manuscript. We feel that the changes we have made draw attention to these concerns, and point to future work that could be done to improve the representation of these complex biosphere-atmosphere exchange processes in GEOS-Chem.

**L 115 – 124: Gaseous dry deposition follows Wang et al. (1998) and Wesely (1989), while particle deposition follows Zhang et al. (2001). In GEOS-Chem, the surface exchange modules are unidirectional (which implies that the effects of bidirectional exchanges of trace gases are not explicitly**

**modelled). In certain regions for which the CEDS inventory scales the calculated emissions to a regional inventory, the extent of accounting for bidirectional exchange may depend on the underlying assumptions in the regional inventory modeling. For example, agricultural ammonia emissions from NEI for the United States includes considering bidirectional ammonia exchange modeling from the Community Multiscale Air Quality Modeling System (CMAQ)** (US EPA, 2018) **. However, we cannot comment with certainty how this is treated elsewhere across the globe, so we assume that neglecting bidirectional exchange of ammonia (and other species for which an atmospheric compensation point may exist) introduces some uncertainty in our simulation (which we discuss in a subsequent section).**

**L 136 – 144: For this study, "agricultural emissions" specifically refer to $NO_x$ and $NH_3$ emitted from fertilizer application and manure management, which correspond directly to agricultural nitrogen input. We do not consider the changes in agricultural of other trace species (e.g. $CH_4$, $SO_2$, CO).** **For simplicity, we assume that agricultural emissions from fertilizer application in CEDS represent "above canopy" emissions to the atmosphere (instead of making assumptions about the implicit treatment of canopy reduction over each region). We note that the fertilizer emissions of represent only a fraction of the total agricultural $NH_3$ emissions we are considering here (e.g. which also include livestock operation), so that uncertainty in a canopy reduction will only affect a fraction of the total. Likewise, fertilizer $NO_x$ emissions are small compared to the total soil $NO_x$ emissions (for which canopy reduction is accounted for online in the Hudman et al. (2012) parameterization).**

**L 228 – 230: Figure 3b shows the changes in annual mean soil NO emission due to LULCC, which represent the change in soil emission driven purely by LAI** **(which can also affect canopy uptake) and land cover** **changes (which affects both biome-based emission factor and canopy uptake) (i.e. without considering the changes in nitrogen input)…**

**L 491 – 506:** **Agricultural $NO_x$ and $NH_3$ emissions estimates also carry large uncertainty due their biological nature and resulting dependence on environmental conditions, which are not explicitly considered in the construction of bottom-up anthropogenic emission inventories (Crippa et al., 2018; Hoesly et al., 2018). Bidirectional exchanges of $NO_2$** (Breuninger et al., 2013; Chaparro-Suarez et al., 2011; Lerdau et al., 2000) **and $NH_3$** (Bash et al., 2013; Massad et al., 2010; Wichink Kruit et al., 2012; Zhang et al., 2010)**are not explicitly modelled (although in some regions may be implicitly accounted for in the regional scaling performed by CEDS), which introduces some uncertainty in the accuracy of surface flux modelling. Zhu et al. (2015) implemented a bi-directional $NH_3$ exchange model in GEOS-Chem, and found no substantial improvement with observations in the modelled $NH_3$ concentration, $NH_4^+$ wet deposition and nitrate aerosol concentration compared to the default GEOS-Chem uni-directional exchange framework. This indicates the uni-directional framework may still be sufficiently accurate in simulating global air quality comparing to bi-directional framework, which requires more observations to properly parameterize at global scale. In the case of $NO_2$, we make the assumption that in most regions we are interested in (fig. S9), the ambient concentrations of $NO_2$ exceed an ecosystem compensation point (0.05-0.6 ppb) (e.g. Breuninger et al. 2013) so that we can assume deposition would dominate. The simplistic representation of dry deposition in general, particularly the lack of dependence of stomatal conductance on atmospheric and soil water content, may not adequately capture the effects of LULCC, as biomes can have differential responses to meteorological and hydrological conditions. The inherent…**

*Lines 333-334: Here there is an apparent flaw: "In contrast, modelled surface ozone increases by up to 1.2 ppbv further south, where strong increases in LAI lead to largely increases vd"; O3 increasing due to enhanced dry deposition? It should also read as "lead to large increases in vd" and what is large? Give a percentage or the absolute numbers.*

Response: We thank the editor for pointing out our mistake. We have made the following correction:

**L 353 – 354: In contrast, modelled surface ozone  decreases by up to 1.2 ppbv further south, where strong increases in LAI lead to largely increases $v_d$ (up to 0.06 m s$^{-1}$).**

*Line 336: "up to 0.6 ppbv of surface ozone increases are simulated, mainly because of the relatively large increase in soil NO emission". This is an example that triggers the question what happened to the effective emissions into the atmosphere; is it indeed purely the changes in the soil NO emissions (due to temperature or moisture effects, or management) and how much an effect is there by changes in the canopy reduction factor due to changes in LAI?*

Response: We thank the editor for raising this interesting question. The relatively strong increase in soil NO emission over West Africa is likely due to the combination of reduced LAI (and therefore lower canopy reduction factor) and cropland expansion. We have made the following changes:

**L 232 – 234: Relatively large increases in soil NO is simulated over western Africa due to both cropland expansion and LAI reduction, which leads to smaller canopy reduction factor and larger emission factor.**

*Lines 428-230; these conclusions are consistent with the findings by Ganzeveld et al. on the small impact of future LULCC and agricultural emissions changes on ozone also due to a number of compensating effects. I think it would be very useful to stress that your findings on contemporary versus future changes in LULCC and agricultural emissions in different modelling systems/approaches come up with such a consistent finding.*

Response: We agree with the editor, that the similarity of conclusion under different timeframe and modelling framework may start to indicate certain generality, which is important to note. We have made the following changes:

**L 447 – 450: We find that the role of LULCC over 1992 to 2014 is regionally significant enough to induce changes in BVOC emissions and dry deposition which affect surface O$_3$, but that the overall effects largely offset each other on the global scale, leading to very small population-weighted $\Delta O_3$, LULCC+agr_emis. This finding with consistent with that of Ganzeveld et al. (2010), even the timeframe of study (2000 – 2050) is different.**

---

## Author Response (AR4)

Dear Handling Editor,

Many thanks for your careful review of our manuscript. We appreciate your time, and value your concerns. We believe we have addressed all your concerns in our revised manuscript. Below, we respond to each of your comments individually, followed by the appropriate changes to our manuscript.

We look forward to your timely response to this revision, and hope that you will agree our manuscript has improved in response to your input.

*I have checked your response to the comments I raised in my editors review and the revisions you included in the manuscript. It appears that overall you appreciated the feedback seeing the revisions. I now accept the paper for publication in ACP after you have handled some minor last issues I found in reading your response and the revision,*

**Comments**

*Line 140: "the fertilizer emissions of represent only.."*

*Line 496: "Zhang et al., 2010) are" (space missing)*

Response: We have made all revisions above as suggested.

*Lines 352:353; " In contrast, modelled surface ozone increases decreases by up to 1.2 ppbv further south, where strong increases in LAI lead to largely increases in vd (up to 0.06 m s-1)"; An increase of 0.06 m s-1 is an increase of 6 cm s-1. That is indeed a large Vd given that the maximum for ozone is ~2 cm s-1.... If it is a typo and should be 0.06 cm s-1 than it is not a large change; a change of 0.6 cm s-1 would be indeed a large change. So what is it at the end?*

Response: We thank the editor for pointing out our mistake. We have made the following corrections:

**L 352 – 353: In contrast, modelled surface ozone decreases by up to 1.2 ppbv further south, where strong increases in LAI lead to  increases in $v_d$ (up to 0.06 cm s$^{-1}$).**